# A gradient green-beard gene in fission yeast

Zhiwei Wu [ID] & Guan-Zhu Han [ID] [✉]

## Abstract

The social behaviors of microbes provide unique opportunities for testing social evolution theories. How can altruistic behaviors arise by natural selection is a central challenge in biology. Green-beard effect has been proposed as a basic mechanism for the evolution of altruistic behaviors. Yet, green-beard genes are generally thought to be rare. Here, we find that the *Schizosaccharomyces pombe gsf2* gene mediates flocculation-like aggregation, and flocculation is triggered by acid stresses. *gsf2*-expressing cells preferentially adhere to each other. The expression of *gsf2* is costly, but *gsf2*-expressing cells preferentially adhere to each other and protect each other from external stress. Gsf2 is highly variable in natural populations, likely contributing to different flocculation intensity. These findings suggest that *gsf2* is a gradient green-beard gene that drives the altruism among *gsf2* carriers. Moreover, we find that *gsf2* is a new gene that originated very recently. Our results provide insights into the origin and evolution of green-beard genes.

**Keywords** Green-beard Gene; Altruistic Behaviors; Flocculation; Fission Yeast
**Subject Categories** Evolution & Ecology; Microbiology, Virology & Host Pathogen Interaction

## Introduction

Darwin (Darwin, 1895) argued that natural selection favors individuals with greater reproductive success "in the struggle for life". After decades, Fisher, Haldane, and Wright synthesized Mendel's theory of heredity and Darwin's theory of natural selection (Provine, 1971), predicating that the frequency of alleles associated with greater individual fitness increases under natural selection. However, the theoretical framework of "individual-centered" natural selection could not account for the evolution of social behaviors that are costly to the individuals that perform them, such as altruism (Hamilton, 1964a; West and Gardner, 2010). How altruistic behaviors arise by natural selection had long been a fundamental challenge in biology.

The theory of kin selection, formulated based on inclusive fitness, is a major theoretical attempt to explain the evolution of altruism and of social behaviors in general (Bourke, 2011; Hamilton, 1964a, b; Wilson, 2000). The theory shows that an altruistic trait will be favored by natural selection when $rb - c > 0$, where $b$ is the fitness benefit received by the recipient, $c$ is the fitness cost of the trait to the actor, and $r$ is their genetic relatedness. Crucially, Hamilton (Hamilton, 1964b) noted that a gene (known as the green-beard gene) that encodes an altruistic trait can favored by kin selection even when the actor and the recipient are not genealogical kin, provided that the gene gives rise to a conspicuous label (for example, greenbeard), confers perception of the label, and behaves altruistically towards bearers of the label (Hamilton, 1964b). Therefore, a green-beard gene possesses three necessary properties, namely label, recognition, and response (Haig, 1996; Hamilton, 1964a; West and Gardner, 2010). The principle of genes appearing to "recognize" copies of themselves in other individuals has been dubbed the "green-beard effect" (Dawkins 1982; Dawkins 2016).

Green-beard gene has generally been thought to be an inherent improbability, because "it is not very probable that one and the same gene would produce both the right label and the right sort of altruism" and cheaters (falsebeards) that display the greenbeard label without performing the altruistic behavior can easily invade the population (Dawkins, 2016; Gardner and West, 2010; Hamilton, 1964b). Yet, the list of proposed green-beard genes is growing (Madgwick et al, 2019; West and Gardner, 2010). For instance, in the red imported fire ant *Solenopsis invicta*, the allele $b$ at the locus *Gp-9* preferentially induces workers bearing the allele to discriminate and execute queens that do not bear it (Keller and Ross, 1998). In the slime mold *Dictyostelium discoideum*, the *csA* gene that encodes a cell adhesion protein promotes cells to stream into an aggregation that differentiates into spores and a stalk made up of cells that altruistically die (Queller et al, 2003).

The complex social behaviors of microbes offer unique opportunities for testing social evolution theories (Strassmann et al, 2011; West et al, 2006). The cells of the budding yeast *Saccharomyces cerevisiae* possess remarkable capability to adhere to surfaces (Verstrepen et al, 2003; Verstrepen and Klis, 2006; Verstrepen et al, 2004). Toward the end of fermentation, the cells of *S. cerevisiae* adhere to each other, forming flocs consisting of hundreds or thousands of cells, and the cell-to-cell adhesion is often known as flocculation. The surface protein FLO1 mediates flocculation of *S. cerevisiae*, protecting the *FLO1*-expressing cells from diverse stresses. The *FLO1* gene has been claimed to be a green-beard gene (Smukalla et al, 2008). Yet, it remains unclear whether flocculation-like aggregation is a common altruistic behavior in microbes.

Flocculation-like aggregation has also been observed in the fission yeast *Schizosaccharomyces pombe*. The *FLO1* gene is not

College of Life Sciences, Nanjing Normal University, Nanjing, Jiangsu 210023, China. ✉E-mail: guanzhu@njnu.edu.cn

present in *S. pombe*, but the *gsf2* gene, which encodes a cell-surface protein binding to galactose chains on the cell surface, is responsible for non-sexual flocculation in *S. pombe* (Kwon et al, 2012; Matsuzawa et al, 2013; Matsuzawa et al, 2011; Matsuzawa et al, 2012). The expression of *gsf2* is negatively regulated by the transcription factor Gsf1, which contains an N-terminal zinc-finger DNA-binding domain (Matsuzawa et al, 2013). Here, we investigated the function and evolution of *gsf2* in *S. pombe*. Our results suggest that flocculation is an altruistic behavior in *S. pombe*, and *gsf2* is a green-beard gene. Moreover, *gsf2* originated recently in evolution and exhibits high variation, implying that gradient greenbeards are maintained in the natural population.

## Results

### The role of *gsf2* in flocculation of *S. pombe*

When we exposed a laboratory stock of *S. pombe* 972h- (designated the WT strain hereafter) to acetic acid challenge, we observed flocculation-like aggregation. We hypothesize that acetic acid triggers the expression of the *gsf2* gene that was known to be responsible for flocculation in *S. pombe* (Kwon et al, 2012; Matsuzawa et al, 2013; Matsuzawa et al, 2011; Matsuzawa et al, 2012). Due to sequence complexity of *gsf2*, we used the long-read sequencing technology to sequence the genome of the WT strain used in this study. The assembled *gsf2* of the WT strain encodes a protein with 2343 amino acids (aa), whereas the Gsf2 protein in the *S. pombe* reference genome is 1563 aa in length (Fig. 1A). Gsf2 proteins possess four repeat motifs, designated A to D. Motifs A and B were defined previously (Matsuzawa et al, 2011), and motifs C and D were newly defined in this study based on our sequence analyses. The difference between WT and reference Gsf2 proteins lies in the number of repeat motif A (6 in the reference Gsf2 vs. 16 in the WT strain Gsf2) (Fig. 1A). When the WT strain cells were subjected to acetic acid challenge, we observed strong flocculation over time (Fig. 1B; Appendix Fig. S1A–C for replicates). The expression of *gsf2* was significantly upregulated after 2 h of acetic acid challenge (Fig. 1C). However, the expression level of *gsf2* gradually decreased during the acetic acid challenge (Fig. 1C).

To further support our hypothesis that acetic acid induces the expression of the *gsf2* gene and subsequently induces cellular flocculation, we knocked out the *gsf2* gene, generating the *gsf2Δ* strain (Fig. 1D). When the *gsf2Δ* strain was challenged using acetic acid, we did not observe strong flocculation (Fig. 1F). We also replaced the 5'UTR of the *gsf2* gene by the thiamine repressive promoter nmt41, generating a *gsf2* inducible expression strain *gsf2IE* (Fig. 1E). YE medium contains thiamine, in which *gsf2* expression is repressed, whereas the expression of *gsf2* can be induced in thiamine-free EMM. When the expression of *gsf2* was induced in EMM (Fig. 1G), the *gsf2IE* strain showed strong flocculation (Fig. 1F). Together, these results indicate that acetic acid potentially induces cell flocculation through the upregulation of *gsf2* expression.

### Acid stresses trigger flocculation

To identify the potential triggers of flocculation, we challenged the WT strain cells with diverse stresses, including oxidative stress (cultured by adding hydrogen peroxide), hypoxia stress (Lee et al, 2007) (cultured by adding CoCl$_2$), heavy metal stress (cultured by adding CdCl$_2$), hyperthermal stress (cultured in 37 °C), and acid stresses (cultured by adding acetic acid, hydrochloric acid, nitric acid, and oxalic acid). When the WT cells were challenged by hyperthermal, oxidative, heavy metal, and hypoxia stresses, no strong flocculation was observed (Appendix Fig. S2). However, when the WT strain cells were subjected to four acid stresses (acetic acid, hydrochloric acid, nitric acid, and oxalic acid), we observed that the WT strain cells formed flocs (Fig. 2B; Appendix Fig. S3 for replicates). When the WT strain cells were cultured by adding both acids and galactose that can competitively inhibit the binding of Gsf2 and the galactose on cell walls, no strong flocculation was observed (Fig. 2B; Appendix Fig. S3 for replicates). Gsf2 was significantly upregulated when challenged by the four acids or when cultured by adding acids and galactose (Fig. 2A). These results indicate that *gsf2*-induced flocculation is likely to be triggered specifically by the environment cue of hydrogen ion concentration, although other environmental cues might exist.

### Expression of *gsf2* is costly

The expression of *gsf2* is suppressed by *gsf1* (Matsuzawa et al, 2013). To avoid the effect of *gsf1* and directly explore the effect of *gsf2*, we compared *gsf2* knockout strain *gsf2Δ* and *gsf2* inducible expression strain *gsf2IE* in the following study. We explored the effect of *gsf2* expression on the fitness of *S. pombe* in normal culture condition. The *gsf2Δ* strain was planktonic, whereas the *gsf2IE* strain, when the expression of *gsf2* was induced, formed flocs (Fig. 1F,G). We found that the flocculent *gsf2IE* strain, when the *gsf2* expression was induced, grew more slowly than the planktonic *gsf2Δ* strain (Fig. 3A). Moreover, the maximum growth rate of the *gsf2IE* strain is significantly lower than that of the *gsf2Δ* strain ($p = 0.0052$, Student's t-tests) (Fig. 3A). It follows that the expression of *gsf2* is costly, and *gsf2*-induced flocculation impairs *S. pombe* fitness in normal culture condition.

### *gsf2Δ* imposes a fitness cost on *gsf2IE*

Given some *gsf2Δ* cells can enter the flocs, they might be cheaters. We explored whether *gsf2Δ* acts as a "cheater" by imposing a fitness cost on the *gsf2IE* strains or simply as a "beneficiary" that does not. We compared the density (assessed by CFU counts) of *gsf2IE* in the monoculture of *gsf2IE* and in the mixed culture of *gsf2Δ* and *gsf2IE* under 0.25% acetic acid stress. If *gsf2Δ* imposes no cost on the fitness of *gsf2IE*, the ratio of *gsf2IE* density in the monoculture to that in the mixed culture will be ~50%. We collected samples based on the growth curve under acetic acid stress, at the end of the logarithmic phase (36 h) and during the stationary phase (60 h and 72 h) (Fig. 3B). We observed significant reduction in the density of *gsf2IE* at 36 h ($p = 0.0064$, Student's t-test), 60 h ($p = 0.0002$, Student's t-test), and 72 h ($p = 0.0034$, Student's t-test) (Fig. 3C). These results indicate that *gsf2Δ* is not a beneficiary, but is a cheater that directly harms the fitness of *gsf2IE*.

### Flocculation escapes cheater exploitation

We also investigated whether flocculation can escape exploitation of the *gsf2Δ* cheaters. We mixed the *gsf2IE* cells and the *gsf2Δ* cells

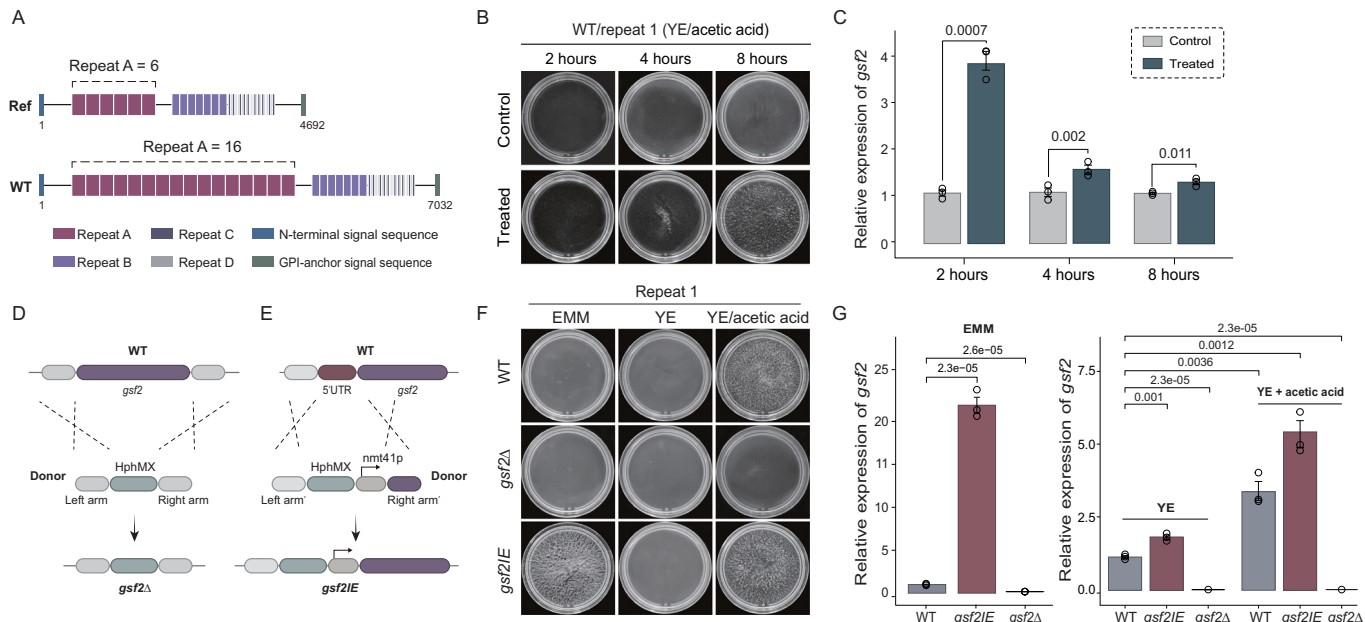

**Figure 1. *gsf2* confers strong flocculation in *S. pombe*.**

(A) The architecture of Gsf2 proteins from the WT strain and the reference genomes. The number indicates the length of Gsf2 proteins. Four repeat motifs are shown. (B) The phenotypes of the WT cells after 2 h, 4 h, and 8 h with and without 0.1% acetic acid treatment. A representative image from three independent biological replicates is shown for each time point and condition. The other replicates are provided in Appendix Fig. S1A– C. (C) Relative expression levels of *gsf2*, measured by quantitative RT-PCR (qPCR), following treatment with or without 0.1% acetic acid for 2, 4, and 8 h. Error bars represent the mean ± SD of three biological replicates. Significance was determined by Student's t-test. (D, E) Schematic diagram for constructing *gsf2* knockout strain *gsf2Δ* and *gsf2* inducible expression strain *gsf2IE*. (F) The phenotypes of the yeast strains WT, *gsf2IE*, and *gsf2Δ* under different culture condition, including EMM media, YE media, and YE media with 0.1% acetate acid. A representative image from three independent biological replicates is shown for each condition. The other replicates were provided in Appendix Fig. S1D–F. (G) Relative expression levels of *gsf2* in the yeast strains WT, *gsf2IE*, and *gsf2Δ* cultured in EMM media, YE media, or YE media with 0.1% acetic acid. Error bars represent the mean ± SD of three biological replicates. Significance was determined by Student's t-test. Source data are available online for this figure.

in a ~ 1:1 ratio and induced the expression of *gsf2*. The flocculant and planktonic cells were then separated and counted serially (Fig. 3D). When the expression of *gsf2* was not induced, the ratio of the *gsf2IE* cells and the *gsf2Δ* cells fluctuated around 1:1 (Fig. 3E). However, when the expression of *gsf2* and flocculation was induced, the *gsf2Δ* and the *gsf2IE* cells were significantly enriched in planktonic and floc fraction, respectively (Fig. 3F,G). The percentage of the *gsf2IE* cells reached more than 80% in flocs and dropped to less than 20% in planktonic fraction after 36 h (Fig. 3F,G). We also mixed the *gsf2IE* cells and the *gsf2Δ* cells in a ~ 100:1 ratio to model a situation where cooperative *gsf2IE* cells are challenged by a low-frequency invasion of *gsf2Δ* cheaters, and induced the expression of *gsf2*. We cultured the cell mixture under both normal condition and 0.25% acetic acid stress, and monitored the proportion of the *gsf2Δ* cells every 24 h (Fig. 3H). Under normal culture conditions, the proportion of *gsf2Δ* consistently fluctuated around 1%. In contrast, under acetic acid stress, the proportion of *gsf2Δ* significantly decreased (Fig. 3H). These results suggest that the aggregation among *gsf2*-expressing cells can alleviate the exploitation of the *gsf2Δ* cheater cells.

## *gsf2*-induced flocculation protects cells against stresses

To investigate whether *gsf2*-induced flocculation protects yeast cells against stresses, we first tested the survival rates of the WT and the *gsf2Δ* strains. Flocculation was induced by 0.1% acetic acid in YE medium (Fig. 3I), and subsequently, we measured cell survival rates under a higher concentration of acetic acid stress (1% acetic acid) using standard colony-forming unit (CFU) counts. We found that under acetic acid stress, the survival rate of the WT strain was significantly higher than that of the *gsf2Δ* strain ($p = 0.017$, Student's t-tests). Furthermore, we determined the survival rates of *gsf2Δ* and *gsf2IE* strains under acid stress (cultured by adding acetic acid), oxidative stress (cultured by adding hydrogen peroxide), and ethanol stress. Flocculation in the *gsf2IE* strain was induced by cultivation in EMM medium (Fig. 3K). We found that the cell survival rate of the *gsf2IE* strain was also significantly higher than that of the *gsf2Δ* strain under all three stresses ($p = 0.0038$ for the addition of acetic acid; $p = 0.019$ for the addition of hydrogen peroxide; and $p = 0.012$ for the addition of ethanol; Student's t-tests) (Fig. 3J,L; Appendix Fig. S4). Therefore, these results suggest that *gsf2*-induced flocculation enables yeast cells to cope with diverse stresses.

To assess the trade-off between the cost of induced *gsf2* expression (Fig. 3A) and its protective benefit under stress, we directly compared the growth of flocculating and non-flocculating cells in the presence of acetic acid. We monitored the growth of the *gsf2Δ* and *gsf2IE* strains under acetic acid stress. Strikingly, under this condition, the flocculating *gsf2IE* strain exhibited a significantly higher maximum growth rate and reached a greater final cell density than the planktonic *gsf2Δ* strain ($p = 0.013$, Student's t-test) (Fig. EV1). This result demonstrates that the protective benefit of

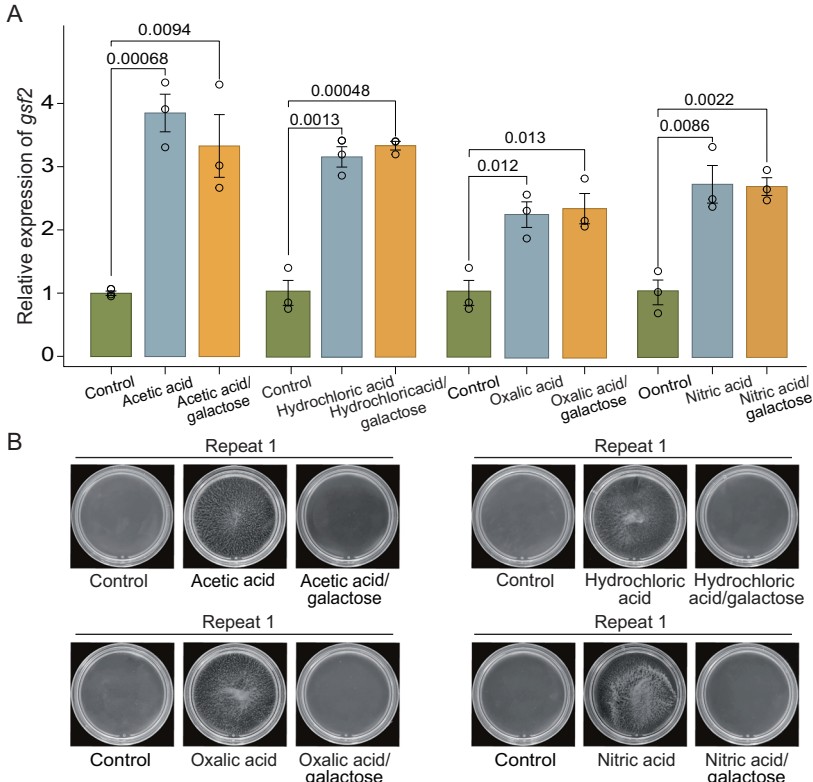

**Figure 2. Acids trigger *gsf2* expression and flocculation.**

(A) The relative expression levels of *gsf2* in the WT strains grown in YE media, YE media supplemented with 0.1% acetic acid, 5 mM hydrochloric acid, 2.5 mM oxalic acid, 4 mM nitric acid and in media containing acid and galactose. Error bars represent the mean ± SD of three biological replicates. Significance was determined by Student's t-test. (B) The phenotypes of the WT cells without and with treatment with 0.1% acetic acid, 5 mM hydrochloric acid, 2.5 mM oxalic acid, and 4 mM nitric acid and with adding acid and galactose. A representative image from three independent biological replicates is shown for each condition. The other replicates were provided in Appendix Fig. S3A–D Source data are available online for this figure.

*gsf2*-mediated flocculation outweighs its inherent cost during acid stress.

### *gsf2*-expressing cells preferentially adhere to each other

We investigated the spatial arrangement of mixed cell populations using confocal microscopy. WT cells (labeled with mCherry) and *gsf2Δ* cells (labeled with GFP) were co-cultured at an approximate 1:1 ratio under 0.1% acetic acid induced *gsf2* expression (Fig. 4A; Appendix Fig. S5A for more views). The results demonstrated that most WT cells aggregated, forming flocs, and that many *gsf2Δ* cells were incorporated into the aggregates and intermixed with the WT cells. For the co-culture of *gsf2IE* (mCherry-labeled) and *gsf2Δ* (GFP-labeled) cells inoculated at an equal ratio (Fig. 4B; Appendix Fig. S5B for more views), a large fraction of *gsf2Δ* cells were planktonic and external to the flocs suggesting that the formation of flocs is driven by the preferential adhesion of *gsf2*-expressing cells to one another.

### *gsf2* is highly variable in natural populations

We analyzed the diversity of Gsf2 proteins in 36 globally sampled strains of *S. pombe* that were sequenced using long-read sequencing approaches (Tusso et al, 2022) (Appendix Table S1).

The four repeat motifs of Gsf2, namely A (78 aa), B (44 aa), C (14 aa), and D (8 aa), are well conserved (Fig. 5B). Premature stop codons were identified in the *gsf2* genes of eight *S. pombe* strains (JB854, JB864, JB953, JB1197, DY39827, DY34373, JB1205, and JB1206), indicating that these *gsf2* genes are likely to be pseudogenes. These strains were therefore excluded from the following analyses. Gsf2 proteins of the remaining strains possess highly variable number of A (5 to 35), B (2 to 8), C (4 to 38), and D (7 to 147) repeats (Fig. 5A,C). Together, our results show that Gsf2 displays high polymorphism in terms of both sequence and presence/absence in natural populations. Interestingly, synteny analysis revealed no homologous or orthologous gene of *gsf2* in species related to *S. pombe* (Appendix Fig. S6). Moreover, we performed tBLASTn search using the individual repetitive motifs (A–C; motif D was too short) as queries against the fungus genomes and NCBI non-redundant nucleotide database. No significant homologous sequences were detected outside of *S. pombe*. Therefore, these results suggest that *gsf2* is likely a novel gene that originated very recently.

### Gsf2 variation affects flocculation intensity

We further investigated whether *gsf2* variation affects the intensity of flocculation. We compared the flocculation intensity between

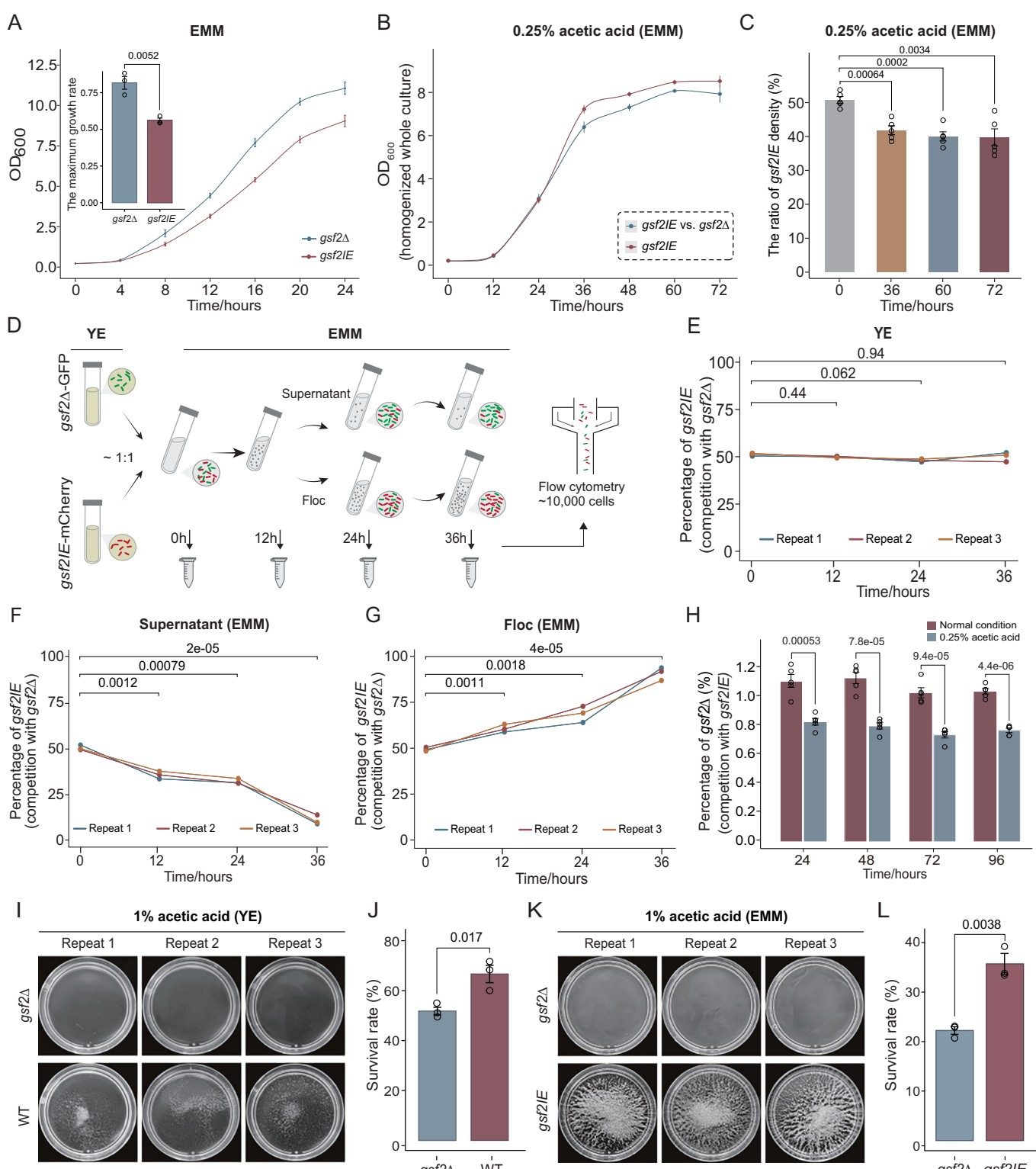

WT and the WT[A] strain, which is also derived from the laboratory strain 972 h[-]. We sequenced the genome of WT[A] using long-read sequencing, and found that the WT[A] strain encodes a Gsf2 protein with 3747 aa. The difference between WT and WT[A] Gsf2 proteins lies in the number of repeat motif A (16 in the WT strain Gsf2 vs.

34 in the WT[A] strain Gsf2) (Fig. 6A). In the WT[A] strain, we replaced the 5'UTR of the *gsf2* gene with the nmt41 promoter, generating the *gsf2[A]IE* strain. When the expression of *gsf2* was not induced, no significant difference in growth was observed between *gsf2IE* and *gsf2[A]IE* (Fig. 6B). However, upon induction of *gsf2*

◄   **Figure 3.   Fitness effect of *gsf2* expression.**

(A) The growth curves of *gsf2IE*, and *gsf2Δ* in EMM media, when the expression of *gsf2* was induced. The maximum growth rates for both strains were also shown. Error bars represent the mean ± SD of three biological replicates. Significance was determined using Student's t-test. (B) Growth curves of the monoculture of *gsf2IE* and the 1:1 mixed culture of *gsf2IE* and *gsf2Δ* in EMM medium supplemented with 0.25% acetic acid. Results are mean ± SD, $n = 3$ biological replicates. (C) The ratio of *gsf2IE* strain density in competitive co-culture over time. The *gsf2IE* and *gsf2Δ* strains were mixed at a 1:1 ratio and co-cultured in EMM medium supplemented with 0.25% acetic acid. The relative abundance of the *gsf2IE* strain within the total population was determined at 36, 60, and 72 h. Data are presented as the mean ± SD of five biological replicates. Significance was determined by Student's t-test. (D) Workflow for monitoring the frequency of the *gsf2IE* (labeled with mCherry) and the *gsf2Δ* (labeled with GFP) cells in the planktonic and flocculent fraction. The *gsf2IE* and the *gsf2Δ* cells were mixed in a ~1:1 ratio and were cultured overnight in YE media. The cultures were then transferred to EMM media to induce the expression of *gsf2*. The supernatant and the flocculent fractions were separated every 12 h. The number of the *gsf2IE* and the *gsf2Δ* cells were measured using flow cytometry. (E) The proportion of *gsf2IE* over time when both the *gsf2IE* and the *gsf2Δ* cells were cultured in YE media and the expression of *gsf2* was not induced. Significance was determined using Student's t-test. (F, G) The proportion of *gsf2IE* in the supernatant and the flocculent fractions over time. Significance was determined by Student's t-test. (H) The proportion of *gsf2Δ* strains in a mixed culture of *gsf2IE* and *gsf2Δ* at a ratio of 100:1 was measured every 24 h by flow cytometry under normal condition or 0.25% acetic acid challenge. Data are presented as the mean ± SD of five biological replicates. Significance was determined using Student's t-test. (I) The flocculation status of WT and *gsf2Δ* in YE media with 1% acetic acid. (J) The survival rates of WT and *gsf2Δ* in YE media with 1% acetic acid. Data are presented as the mean ± SD of three biological replicates. Significance was determined by Student's t-test. (K) The flocculation status of *gsf2IE* and *gsf2Δ* in EMM media with 1% acetic acid. (L) The survival rates of *gsf2IE* and *gsf2Δ* in EMM media with 1% acetic acid. Data are presented as the mean ± SD of three biological replicates. Significance was determined by Student's t-test. Source data are available online for this figure.

expression, a significant growth difference arose between the two strains ($p = 0.029$, Student's t-tests) (Fig. EV2). We then mixed the *gsf2IE* cells (labeled with mCherry) and the *gsf2^AIE* cells (labeled with GFP) in a ~ 1:1 ratio, induced the expression of *gsf2*, and then counted flocculent cells every 12 h. We found that the fraction of the *gsf2IE* cells significantly increased in flocs over time (Fig. 6C), while their proportion in the supernatant fraction showed a corresponding decrease (Fig. 6D). We also knocked out the *gsf2* gene in the WT^A strain, generating the *gsf2^AΔ* strain. Under acetic acid challenge, no flocculation was observed in either strain, and no significant difference in survival rate was observed between *gsf2Δ* and *gsf2^AΔ* ($p = 0.332$; Student's t-tests) (Fig. 6E,F; Appendix Fig. S7B). However, we found that cell survival rate of *gsf2IE* was significantly higher than that of *gsf2^AIE* under acetic acid challenge ($p = 0.022$; Student's t-tests) (Fig. 6F). Taken together, the results indicate that *gsf2* variation affects the flocculation intensity and the ability of protecting yeast cells from stresses.

## Discussion

The green-beard effect is one of the basic mechanisms accounting for the evolution of altruistic behaviors that benefit the recipients at the fitness expense of the actors (West and Gardner, 2010). A green-beard gene possesses three necessary properties, including label, recognition, and response(Haig, 1996; Hamilton, 1964a; West and Gardner, 2010). *gsf2* encodes a cell-surface protein that displays a label. Except the *gsf2* gene, the *gsf2IE* and the *gsf2Δ* strains share genetic relatedness of 1, because they are essentially derived from the same WT strain. However, *gsf2*-expressing cells "recognize" and preferentially adhere to other *gsf2*-expressing cells. Gsf2 protein mediates the social behavior flocculation, which protect cells from diverse stresses. All these results support that flocculation is an altruistic behavior, and *gsf2* is a green-beard gene in the fission yeast *S. pombe*.

The green-beard effect is vulnerable to the cheating of false-beard individuals that exhibit the greenbeard but do not perform altruistic behavior (Bowen, 2018; Dawkins, 2016). The inherent vulnerability of the green-beard effect is one of the main reasons why green-beard genes have been thought to be rare (Dawkins, 2016; Madgwick et al, 2019). Consistent with the expectation of

social exploitation, we observed that the non-producer strain *gsf2Δ* imposes a direct fitness cost on the producer *gsf2IE* in mixed culture (Fig. 3H), likely through competition for common resources while avoiding the cost of *gsf2* expression, which aligns with general models and empirical examples of cheating (Jiricny et al, 2010). However, *gsf2* codes for all the three properties (label, recognition, and response) in a single gene. Thus, *gsf2*-related falsebeards that break the three properties would be difficult to evolve. The *gsf2Δ* cells, which lack the gene entirely and thus do not produce the label, are not such "false-beards." Instead, they are opportunistic cheaters that can be physically embedded within flocs formed by *gsf2IE* cells by chance (Fig. 4A,B), thereby gaining passive access to the group-beneficial acid protection without contributing to floc formation. We demonstrate that despite imposing a local fitness cost, the *gsf2Δ* cheaters cannot ultimately invade the population. The frequency of *gsf2Δ* cells within flocs decreased steadily over time.

In the natural population of *S. pombe*, *gsf2* displays presence and absence polymorphism. *gsf2* activation is likely triggered specifically by acids, although the possibility that other stress cues trigger the expression of *gsf2* cannot be formally excluded. The mechanisms of acid-induced *gsf2* expression merit further exploration. During stress challenge, the activation of *gsf2* mediates flocculation, leading to stress resistance. In absence of stress, disruptive mutations can occur in *gsf2*, which do not impact the overall fitness of the carrier individuals. Thus, maintenance of the presence and absence polymorphism suggest temporal or geographic variation in selection. Moreover, *gsf2* also exhibit high sequence polymorphism, in particular size variation. The size variation might have arisen through recombination mediated by repeat motifs of *gsf2* (Christiaens et al, 2012; Verstrepen et al, 2005; Verstrepen et al, 2004). We show that *gsf2* variation affects the flocculation intensity and thus probably cooperation intensity. Together, we propose that *gsf2* is a gradient green-beard gene. More work is needed to explore how Gsf2 variation has been maintained in nature.

The list of candidate green-beard genes is ever growing (recently reviewed in (Madgwick et al, 2019; West and Gardner, 2010); see references (Heller et al, 2016) and (Gruenheit et al, 2017) for recent examples). In the budding yeast *S. cerevisiae*, the *FLO1* gene that mediates flocculation has been proposed to be a green-beard gene (Smukalla et al, 2008). In this study, we show that the *gsf2* gene is

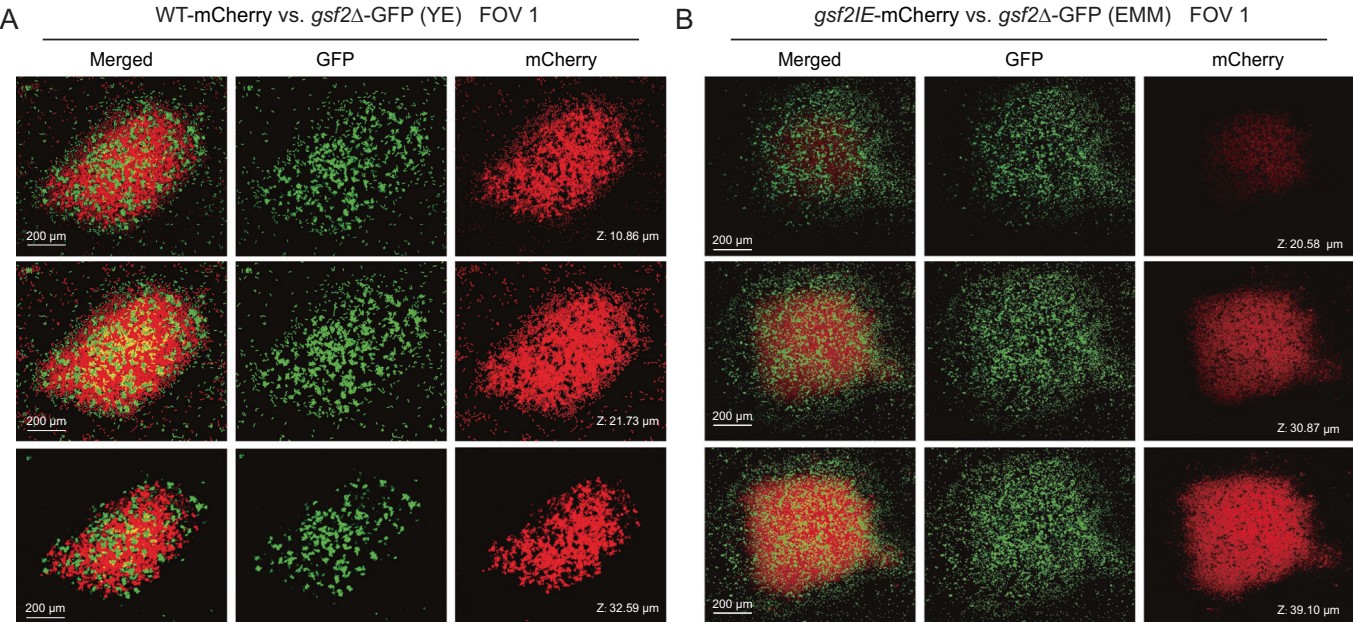

**Figure 4. Spatial arrangement of the mixture of the *gsf2*-expressing and non-expressing cells in the flocculation.**

We performed 3D reconstruction of two flocs using laser confocal microscopy. (A) The floc was formed by an equally mixed population of WT-mCherry and *gsf2Δ*-GFP, induced with 0.1% acetic acid. (B) The floc was formed by an equally mixed population of *gsf2IE* -mCherry and *gsf2Δ*-GFP in EMM medium. The Z value represents the distance of the scanned layer to the bottom of the floc in the Z-dimension. The other views were provided in Appendix Fig. S5.

responsible for flocculation in fission yeast *S. pombe*, and is also a green-beard gene. Both *gsf2* and *FLO1* genes regulate the formation of flocs, but *FLO1*-induced and *gsf2*-induced flocs exhibit difference in morphology and size. For instance, *FLO1*-induced flocs (typically 5–8 mm in diameter (Smukalla et al, 2008)) appear to be much larger than these induced by *gsf2*. Furthermore, the mechanisms of cell recognition and binding of FLO1 and Gsf2 are different: FLO1 recognizes mannose residues, whereas Gsf2 recognizes and binds to galactose residues on extracellular sugar chains (Matsuzawa et al, 2011). Albeit bearing the name of yeast, *S. pombe* and *S. cerevisiae* are distantly related, which diverged from each other around 330 to 420 million years ago (Sipiczki, 2000). *FLO1* and *gsf2* genes share no detectable sequence similarity. Therefore, flocculation represents altruistic behaviors that originated convergently in *S. cerevisiae* and *S. pombe*.

The *gsf2* and the *FLO1* genes belong to adhesins that share the common property of promoting cell adhesion. The aggregation protects internal cells from stress challenge at the sacrifice of peripheral cells. However, the aggregation comes at cost, which can restrict inside cells access to nutrients and oxygen and impede the removal of toxic metabolites. These properties make cell adhesion genes potential candidates for the green beard genes (Haig, 1996; Smukalla et al, 2008). Our study supports Haig's insight that cell adhesion genes could provide real-world examples of green-beard genes (Haig, 1996; Summers and Crespi, 2005). Interestingly, evolutionary analysis shows that *gsf2* is a new gene that originated very recently. Therefore, our study suggests that green-beard genes likely arose repeatedly during the evolution of Life, and green-beard genes might be more common than previously appreciated.

## Methods

**Reagents and tools table**

| Reagent/Resource | Reference or Source | Identifier or Catalog Number |
|---|---|---|
| **Experimental models** | | |
| *Schizosaccharomyces pombe* 972 h- (WT) | Lab stock | N/A |
| *Schizosaccharomyces pombe* 972 h- (WTᴬ) | ATCC | ATCC 24843 |
| **Recombinant DNA** | | |
| pFA6a-HphMX-GFP | Lab stock | N/A |
| pYJ19 | Lab stock | N/A |
| pCMV-HA-mCherry | Addgene | cat#191136 |
| pnmt41-*gsf2* | This study | N/A |
| pnmt41-*gsf2*ᴬ | This study | N/A |
| pnmt41-*gsf2*ᴬ*IE_GFP* | This study | N/A |
| pnmt41-*gsf2IE_RFP* | This study | N/A |
| pnmt41-*gsf2Δ_GFP* | This study | N/A |
| **Oligonucleotides and sequence-based reagents** | | |
| PCR primers | This study | Appendix Table S2 |
| **Chemicals, Enzymes and other reagents** | | |
| HiScript III RT SuperMix | Vazyme | cat#R323-01 |

| Reagent/Resource | Reference or Source | Identifier or Catalog Number |
|---|---|---|
| ChamQTM SYBR qPCR Master Mix | Vazyme | cat#Q331-02 |
| HiScript III 1st Strand cDNA Synthesis Kit | Vazyme | cat#R312-01 |
| ClonExpress II One Step Cloning Kit | Vazyme | cat#C112-01 |
| FastPure Total RNA Isolation Kit V2 | Vazyme | cat#RC112-01 |
| Phanta Max Super-Fidelity DNA Polymerase | Vazyme | cat#P505 |
| 2× Taq Master Mix | Vazyme | cat#P111-01 |
| PCR Product Purification Kit | Vazyme | cat#DC301-01 |
| DpnI | Beyotime | cat#D6257S |
| FastPure Plasmid Mini Kit | Vazyme | cat#DC201 |
| hydrogen peroxide | sigma | cat#7722-84-1 |
| Acetic acid | sigma | cat#64-19-7 |
| D-(+) Galactose | Sangon Biotech | cat#59-23-4 |
| Yeast Extract | Thermo Scientific | cat#LP0021B |
| D-(+) Glucose | Sinopharm | cat#10010518 |
| **Software** | | |
| canu | https://github.com/marbl/canu | |
| NIS-Element Viewer | https://www.microscope.healthcare.nikon.com/en_AOM/products/software/nis-elements/software-resources | |
| tBLASTn | https://blast.ncbi.nlm.nih.gov/ | |
| Snapgene | https://www.snapgene.com/ | |
| FlowJo | https://www.flowjo.com/ | |
| R V4.4.2 | https://cran.r-project.org/src/base/R-4/ | |
| GraphPad Prism | https://www.graphpad.com/features | |
| **Other** | | |
| laser confocal microscope | NIKON | A1 Ti-E-A1R |
| Flow cytometry | BD Biosciences | FACSVerse |
| PacBio Sequel IIe | Pacific Biosciences | |

## Yeast strains and growth conditions

The *S. pombe* WT strain was derived from the laboratory strain 972h- and has been long maintained in our laboratory. The WT[A] strain was the laboratory strain 972h- recently purchased from the ATCC Strain Conservation Center. The *S. pombe* strains were grown in rich media (YE) consisting of 3% glucose and 0.5% yeast extract or in Edinburgh minimum media (EMM) consisting of 3% potassium hydrogen phthalate, 0.22% sodium phosphate, 0.5% ammonium chloride, 2% glucose, vitamins, minerals, and salts at 32 °C with shaking at 200 rpm.

## Yeast strain construction

We used homologous recombination-based methods to generate knockout strains, inducible expression strains, and strains with fluorescent protein tags. To construct the *gsf2* knockout strains (*gsf2Δ* and *gsf2*[A]*Δ*), we cloned fragments of ~500 bp upstream and downstream of *gsf2* using PCR. The HpHMX6 cassette conferring hygromycin resistance was amplified from the plasmid pFA6a-5FLAG-hphMX6. All the fragments were then purified by gel and fused using overlapping PCR. To construct the *gsf2* inducible expression strains (*gsf2IE* and *gsf2*[A]*IE*), the 5'UTR of *gsf2* was replaced with the promoter nmt41 cloned from the plasmid pBMod-Pnmt41. The promoter nmt41 is a thiamine-repressible promoter, and the expression of genes under its control is repressed by the presence of thiamine in the medium. YE medium contains thiamine, in which *gsf2* expression is repressed. However, in thiamine-free EMM, the expression of *gsf2* is induced. To construct fluorescent protein-tagged strains, we cloned fluorescent tags under the control of ADH1 promoter from the plasmids pSR240 or pSH100 and then inserted them into gene-poor regions as described in ref. Wang et al (2021).

## Stress resistance assay

Survival rate under stress challenge was measured by counting colony-forming units (CFUs) as described in ref. Liu et al (2011). Briefly, yeast cells were initially cultured overnight in YE liquid medium at 32 °C with shaking at 200 rpm. For the treatment of the WT strain, flocculation was induced by the addition of 0.1% acetic acid to the YE medium. Six replicate cultures were prepared in parallel. Following floc formation (approximately 5 h post-induction), three of these replicates were thoroughly vortexed to ensure complete de-flocculation, and the OD$_{600}$ was promptly measured. The other three replicates were subjected to a 2-hour stress treatment with 1% acetic acid for subsequent survival analysis. To maintain synchronicity, the *gsf2Δ* strain was processed under an identical regimen. For the *gsf2IE* population, flocculation was induced by direct inoculation into fresh EMM medium. Subsequently, the cells were exposed to various stress conditions for 2 h, including 1% acetic acid, 25 mM hydrogen peroxide, or 10% ethanol. Following the stress treatments, a cell suspension volume equivalent to an OD$_{600}$ of 0.3 was serially diluted 1000-fold. A 100 μL aliquot of the diluted suspension was then plated onto non-selective YE solid agar. The plates were incubated at 32 °C for 3 days to allow for colony formation. The survival rate was calculated as the ratio of the number of CFUs from the stress-treated samples to that of the untreated control samples.

## Fitness cost assay

To estimate the fitness cost of the *gsf2* expression, we constructed the growth curves of *gsf2IE* and *gsf2Δ*. Both strains were cultured at 32 °C with shaking at 200 rpm, and the initial OD$_{600}$ was adjusted to ~0.2. Samples were taken at 3-hour period over a 24-h period. Samples were resuspended in 2 M galactose solution and vortexed thoroughly to disperse cell aggregates. The OD$_{600}$ was

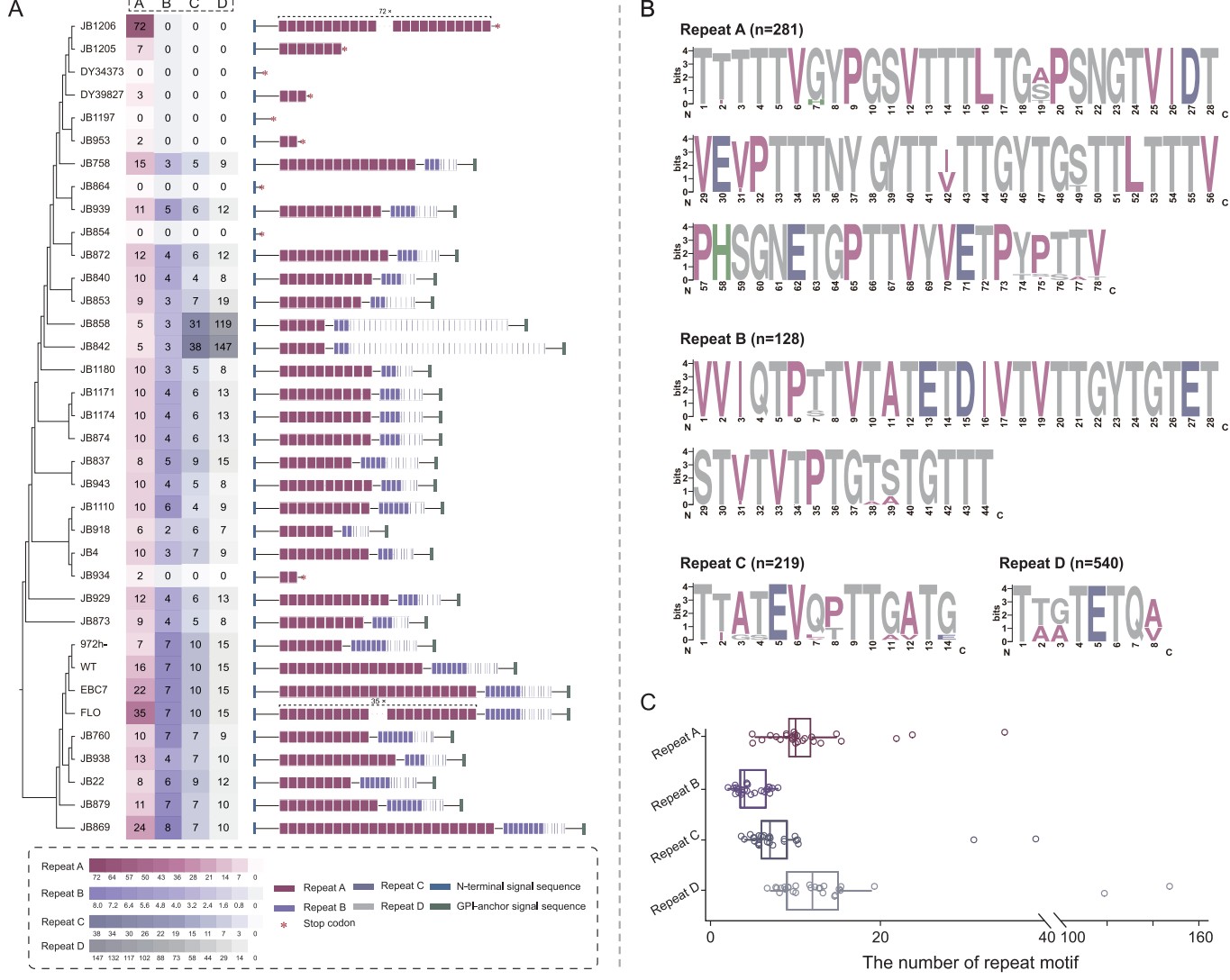

**Figure 5. Diversity of Gsf2 proteins in *S. pombe* strains sampled globally.**

(A) The sequence diversity of Gsf2 proteins from 36 *S. pombe* strains sequenced using long-read sequencing approaches. The number of repeat motifs and the overall architectures are shown near the strain where the Gsf2 were retrieved. Heat map illustrates the number of repeat motifs. Premature stop codons are labeled with red asterisks. The evolutionary relationship among *S. pombe* strains is based on ref. (Tusso et al, 2022). (B) Sequence logo of repeat motifs in Gsf2 proteins. Numbers in parentheses indicate the number of repeat sequences analyzed. (C) The box plot represents variation in the number of repeat motifs among *S. pombe* strains with full-length *gsf2* genes. strains with premature stop codons are not represented. The center line indicates the median; box bounds represent the 25th and 75th percentiles; whiskers show the minimum and maximum values. A total of 27 strains were analyzed (*n* = 27). Source data are available online for this figure.

immediately determined by spectrophotometry (Eppendorf Bio-Photometer 6131). Three replicates were performed for each sample. We used the growth rate of the logarithmic growth period in the growth curve as the maximum growth rate for each strain, which is the ratio of the difference between the end of logarithmic growth and the beginning of logarithmic growth to the time difference.

## Expression level analysis

Yeast strains were grown overnight in 10 ml YE liquid media. The cultures were transferred to fresh YE liquid media, and the initial $OD_{600}$ was adjusted to ~0.2. Yeast strains were treated with 0.1%

acetic acid for 2 h, 4 h, and 8 h. RNA was extracted by Yeast RNA Kit (Vazyme RC411-01). RNA was reverse transcribed into cDNA using HiScript III RT SuperMix (Vazyme R323-01). Reverse transcription quantitative PCR (RT-qPCR) was performed using intercalating dye ChamQTM SYBR qPCR Master Mix (Vazyme Q331-02) to measure the expression level of *Act1* and *gsf2*. The primers were provided in Appendix Table S2. qPCR reactions were performed in 10 μl volume. RNA was extracted three times. Data analysis was performed by StepOne. $C_T$ values were normalized against *Act1* mRNA levels from the same preparation to estimate $\Delta C_T$ values, and the relative changes in gene expression level were estimated using the $2^{-\Delta\Delta C_T}$ method (Livak and Schmittgen, 2001).

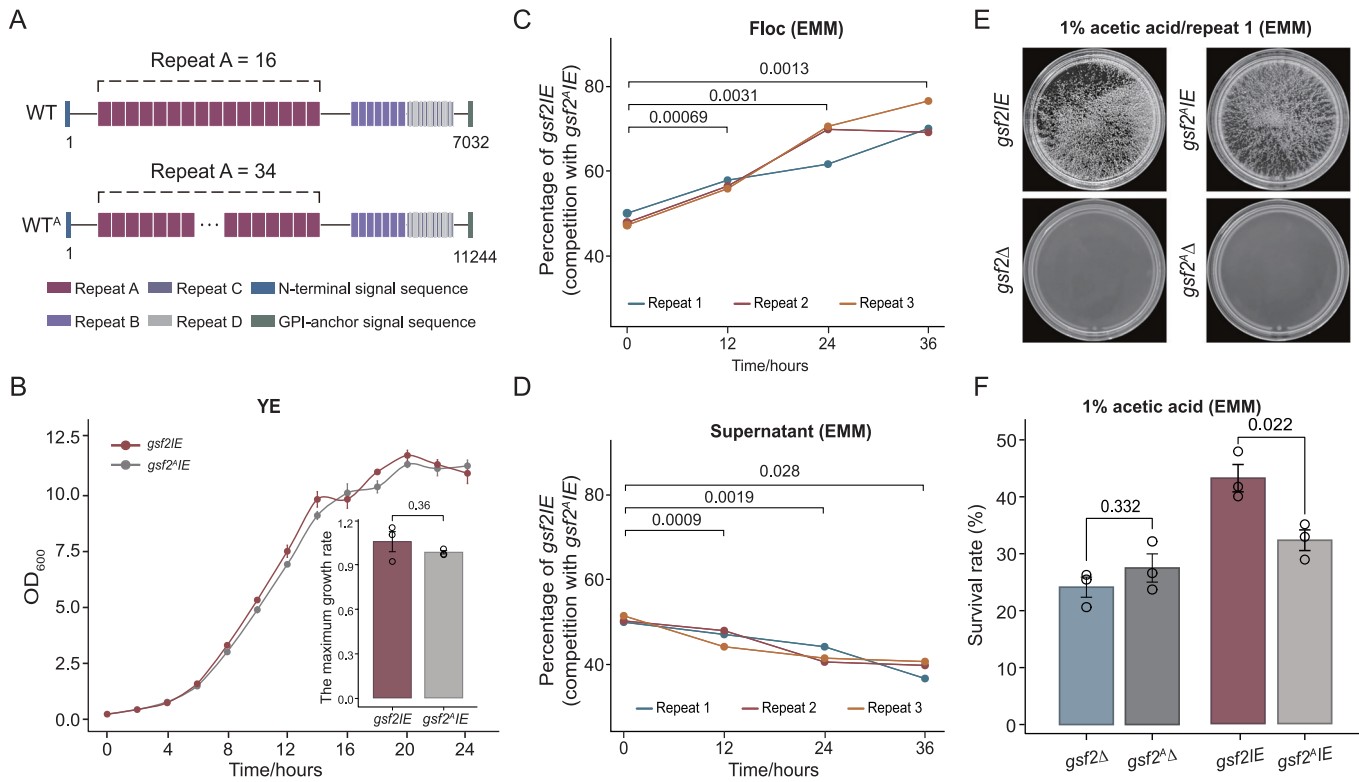

**Figure 6. Gsf2 variation affects flocculation intensity.**

(A) The architecture of Gsf2 proteins from the WT strain and the WT[A] strain genomes. The number indicates the length of Gsf2 proteins. Four repeat motifs are shown. (B) The growth curves and the maximum growth rates of *gsf2IE* and *gsf2[A]IE* strains. $OD_{600}$ was measured every 2 h for 24 h. Data are presented as mean ± standard deviation of three independent experiments. The maximum growth rates for both strains were also shown. Significance was determined using Student's t-test. (C) The proportion of *gsf2IE* cells in the flocculent fractions was measured over time in cultures where *gsf2IE* and *gsf2[A]IE* strains were initially mixed at a 1:1 ratio. $n$ = three biological replicates. Significance was determined using Student's t-test. (D) The proportion of *gsf2IE* cells in the supernatant was measured over time in cultures where *gsf2IE* and *gsf2[A]IE* strains were initially mixed at a 1:1 ratio. $n$ = three biological replicates. Significance was determined using Student's t-test. (E) The flocculation status of *gsf2[A]Δ*, *gsf2Δ*, *gsf2[A]IE* and *gsf2IE* in EMM media with 1% acetic acid. (F) The survival rates of *gsf2[A]Δ*, *gsf2Δ*, *gsf2[A]IE* and *gsf2IE* after treatment of 1% acetic acid. Error bars represent the mean ± SD of three biological replicates. Significance was determined using Student's t-test. Source data are available online for this figure.

## Confocal fluorescence microscopy

We used confocal microscopy to investigate the spatial arrangement of a mixed population of *gsf2*-expressing and non-expressing cells in the flocculation. The yeast strains were first cultured overnight in YE liquid medium at 32 °C with shaking at 200 rpm, and then washed with sterile water. We mixed *gsf2Δ*-GFP and WT-mCherry at an approximate 1:1 ratio and inoculated the mixture into fresh YE liquid medium supplemented with 0.1% acetic acid for 5 h to induce *gsf2* expression and floc formation. Subsequently, this mixture was transferred to a confocal dish, and spatial distribution images were acquired using a NIKON A1 Ti-E-A1R laser confocal microscope equipped with a 10× objective lens. Image processing was performed using NIS-Element Viewer 4.20 software. For the population of *gsf2IE*-mCherry and *gsf2Δ*-GFP mixed in equal proportions, we directly transferred them to fresh EMM medium to induce *gsf2* expression and generate flocculation particles. The subsequent procedures were the same as described above.

## Cheater exploration experiment

*gsf2IE* (labeled with mCherry) and *gsf2Δ* (labeled with GFP) cells were cultured in YE liquid media at 32 °C overnight. Cultures were washed with sterile water and mixed in a ~ 1:1 ratio. Supernatant and flocculated cells were separated and transferred to fresh EMM liquid media every 12 h. The number of *gsf2IE* cells and *gsf2Δ* cells were counted using the FITC channel of the flow cytometry BD FACSVerse. In brief, the supernatant and flocculated cells were washed three times with sterile water, then resuspended in 2 M galactose solution, and vortexed thoroughly to disperse cell aggregates. A total of 10,000 cells from the samples were measured by flow cytometry, and the ratio of *gsf2IE* cells and *gsf2Δ* cells were estimated.

We also mixed the *gsf2IE* cells and the *gsf2Δ* cells in a 100:1 ratio, and $OD_{600}$ was adjusted to 0.1 for *gsf2IE* and 0.001 for *gsf2Δ*. Multiple parallel mixed cultures were performed. At each sampling time point, 20 μL of 2 M galactose was added to three mixed cultures, which were then disaggregated using vortex. The cells

were washed three times with sterile water. A total of 10,000 cells from the samples were measured by flow cytometry, and the ratio of *gsf2IE* cells and *gsf2Δ* cells were estimated.

To assess the effect of *gsf2Δ* on the fitness of *gsf2IE*, a monoculture of *gsf2IE* and a 1:1 mixed culture of *gsf2IE* and *gsf2Δ* were grown in EMM medium with 0.25% acetic acid. For both cultures, the initial $OD_{600}$ was adjusted to 0.2. Multiple parallel cultures were performed. At each sampling time point, 20 μL of 2 M galactose was added to the cultures, which were then vortexed to ensure complete disaggregation of the flocs. The density of *gsf2IE*, which carries a hygromycin resistance marker, was determined by plating on selective media and counting CFUs. The effect of *gsf2Δ* on *gsf2IE* was quantified as the ratio of the *gsf2IE* density in the mixed culture to that in the monoculture.

## Genome sequencing

Yeast cells were grown in 20 ml YE liquid media overnight. Cells were harvested using centrifugation and washed three times with sterile water. The genomic DNA was extracted using a modified glass bead lysis method (Lang et al, 2013). Genomic DNA was sequenced using the Pacific Biosciences (PacBio) long-read sequencing technology provided by Frasergen Biotechnology (Wuhan, China). Canu 2.0 (Koren et al, 2017) was used for de novo genome assembly. QUAST 5.0.2 (Gurevich et al, 2013) was used to evaluate the assembly quality. Sequencing data generated in this study are available at the CNGB Sequence Archive (CNSA) (Guo et al, 2020) (https://db.cngb.org/cnsa) of China National GeneBank DataBase (CNGBdb) with accession number CNP0006508.

## Statistical analyses

Each experiment included at least three independent biological replicates per group ($n = 3$). Blinding was not applicable during data collection and analysis because the primary measurements were obtained using automated instruments, thereby eliminating observer bias. Data are presented as mean ± standard deviation (SD). The normality of data distribution was assessed using the Shapiro-Wilk test. Homogeneity of variances was confirmed using F-test. For comparisons between two groups, unpaired two-tailed Student's t-test was performed when data passed normality and equal variance tests; otherwise, the Mann-Whitney U test was used. Statistical analysis was performed using GraphPad Prism software. A $p$-value < 0.05 was considered statistically significant.

## Data availability

Sequencing data generated in this study are available at the CNGB Sequence Archive (CNSA) (https://db.cngb.org/cnsa) of China National GeneBank DataBase (CNGBdb) with accession number CNP0006508. The confocal microscopy 3D images have been deposited in the BioStudies database (Sarkans et al, 2018), under accession number S-BSST2567.

The source data of this paper are collected in the following database record: biostudies:S-SCDT-10_1038-S44319-026-00748-x.

## Peer review information

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

## Acknowledgements

We thank the Nanjing Normal University Large Instrument Platform for providing support with flow cytometry and confocal microscopy. We also thank Personalbio Biotechnology Company for providing PacBio HiFi sequencing services. This work was supported by the National Natural Science Foundation of China (Grant Nos. 32470652 and 32270684).

## Author contributions

**Zhiwei Wu**: Investigation; Writing—original draft; Writing—review and editing. **Guan-Zhu Han**: Conceptualization; Supervision; Writing—original draft; Writing—review and editing.

Source data underlying figure panels in this paper may have individual authorship assigned. Where available, figure panel/source data authorship is listed in the following database record: biostudies:S-SCDT-10_1038-S44319-026-00748-x.

## Disclosure and competing interests statement

The authors declare no competing interests.

# Expanded View Figures

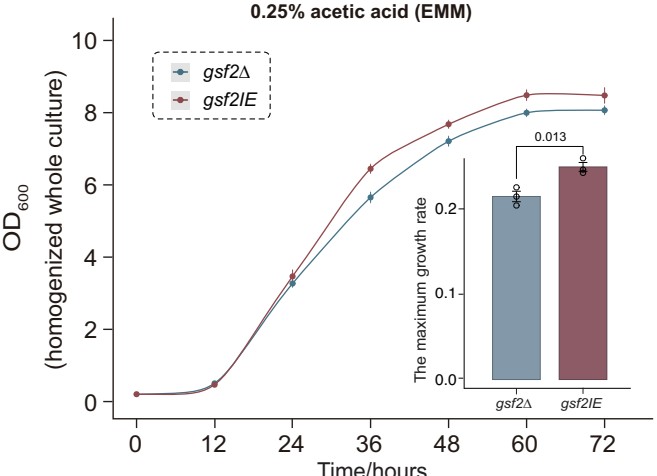

**Figure EV1.  *gsf2* confers a growth advantage under acetic acid stress.**

Growth curves of *gsf2Δ* and *gsf2IE* strains cultivated in EMM medium supplemented with 0.25% acetic acid, under *gsf2*-inducing conditions. The maximum growth rates for both strains are quantified. Error bars represent mean ± SD of three biological replicates. Statistical significance was determined using a two-tailed Student's t-test. Source data are available online for this figure.

                                                                              

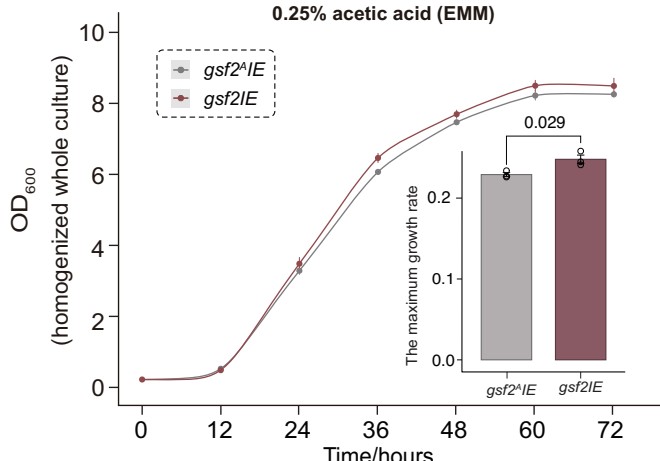

**Figure EV2. The growth dynamics of *gsf2^AIE* and *gsf2IE* strains under acetic acid stress.**

The maximum growth rates for both strains were also shown. Error bars represent the mean ± SD of three biological replicates. Significance was determined using Student's t-test. Source data are available online for this figure.

