## [Peer Review File · EMBO Reports]

A Gradient Green-beard Gene in Fission Yeast

Zhiwei Wu and Guan-Zhu Han

Corresponding author(s): Guan-Zhu Han (guanzhu@njnu.edu.cn)

Review Timeline:

Submission Date:	14th Oct 25
Editorial Decision:	10th Dec 25
Revision Received:	6th Jan 26
Editorial Decision:	12th Feb 26
Revision Received:	14th Feb 26
Accepted:	5th Mar 26

Editor: Yehu Moran

Transaction Report:

Dear Dr. Han

Thank you for the submission of your manuscript to EMBO Reports. We have now received the full set of referee reports that are pasted below.

As you will see, the referees acknowledge that the findings are potentially interesting. However, they also raise significant comments and concerns that require your attention.

I would thus like to invite you to revise your manuscript with the understanding that the referee concerns must be fully addressed and their suggestions taken on board. Please address all referee concerns in a complete point-by-point response. Acceptance of the manuscript will depend on a positive outcome of a second round of review. It is EMBO reports policy to allow a single round of major revision only and acceptance or rejection of the manuscript will therefore depend on the completeness of your responses included in the next, final version of the manuscript.

We realize that it is difficult to revise to a specific deadline. In the interest of protecting the conceptual advance provided by the work, we recommend a revision within 3 months (12th Mar 2026). Please discuss the revision progress ahead of this time with the editor if you require more time to complete the revisions.

- 1) A data availability section providing access to data deposited in public databases is missing. If you have not deposited any data, please add a sentence to the data availability section that explains that.
- 2) Your manuscript contains statistics and error bars based on $n=2$. Please use scatter blots in these cases. No statistics should be calculated if $n=2$.

When submitting your revised manuscript, please review the submission guidelines (<https://link.springer.com/journal/44319/submission-guidelines#cms-Revised-submissions>) and carefully review the instructions that follow below. Failure to include requested items will delay the evaluation of your revision.

2) individual production quality figure files as .eps, .tif, .jpg (one file per figure). See <https://media.springernature.com/original/springer-cms/rest/v1/content/27825798/data/v1> for more info on how to prepare your figures.

3) We replaced Supplementary Information with Expanded View (EV) Figures and Tables that are collapsible/expandable online. A maximum of 5 EV Figures can be typeset. EV Figures should be cited as 'Figure EV1, Figure EV2' etc... in the text and their respective legends should be included in the main text after the legends of regular figures.

- For the figures that you do NOT wish to display as Expanded View figures, they should be bundled together with their legends in a single PDF file called *Appendix*, which should start with a short Table of Content. Appendix figures should be referred to in the main text as: "Appendix Figure S1, Appendix Figure S2" etc. See 'Expanded View' section of the submission guidelines.

5) a complete author checklist, which you can download from our author guidelines. Please insert information in the checklist that is also reflected in the manuscript. The completed author checklist will also be part of the RPF.

6) Please note that all corresponding authors are required to supply an ORCID ID for their name upon submission of a revised manuscript (<<https://orcid.org/>>), which can be linked to your profile in the manuscript submission system.

7) Before submitting your revision, primary datasets produced in this study need to be deposited in an appropriate public database. Please remember to provide a reviewer password if the datasets are not yet public. The accession numbers and database should be listed in a formal "Data Availability" section placed after Methods. Please note that the Data Availability Section is restricted to new primary data that are part of this study. * Note - All links should resolve to a page where the data can be accessed. *

9) Our journal also encourages inclusion of *data citations in the reference list* to directly cite datasets that were re-used and obtained from public databases. Data citations in the article text are distinct from normal bibliographical citations and should directly link to the database records from which the data can be accessed. In the main text, data citations are formatted as follows: "Data ref: Smith et al, 2001" or "Data ref: NCBI Sequence Read Archive PRJNA342805, 2017". In the Reference list, data citations must be labeled with "[DATASET]". A data reference must provide the database name, accession number/identifiers and a resolvable link to the landing page from which the data can be accessed at the end of the reference.

10) Regarding data quantification, the following points must be specified in each figure legend:

- the name of the statistical test used to generate error bars and P values,
- the number (n) of independent experiments (please specify technical or biological replicates) underlying each data point,
- the nature of the bars and error bars (s.d., s.e.m.),
- If the data are obtained from n Program fragment delivered error ``Can't locate object method "less" via package "than" (perhaps you forgot to load "than"?) at //ejpvfs23/sites23b/embo_r_www/letters/embo_decision_revise_and_review.txt line 54.' 2, use scatter blots showing the individual data points.

11) The journal requires a statement specifying whether or not authors have competing interests (defined as all potential or actual interests that could be perceived to influence the presentation or interpretation of an article). In case of competing interests, this must be specified in your disclosure statement.

12) All Materials and Methods need to be described in the main text using our 'Structured Methods' format, which is required for all research articles. According to this format, the Methods section includes a Reagents and Tools Table (listing key reagents, experimental models, software and relevant equipment and including their sources and relevant identifiers) followed by a Methods and Protocols section describing the methods using a step-by-step protocol format. The aim is to facilitate adoption of the methodologies across labs. More information on how to adhere to this format as well as a downloadable template (.docx) for the Reagents and Tools Table can be found in our author guidelines.

I look forward to seeing a revised form of your manuscript when it is ready.

Yours sincerely,

Referee #1:

The authors show that cells treated with acid flocculate, that this flocculation is correlated with *gsf2* expression and that this flocculation is dependent on the presence of *gsf2* by showing that it is absent in a *gsf2* deletion and present when *gsf2* is overexpressed, even in the absence of acetic acid.

The authors then show that cells flocculating due to *gsf2* overexpression show a growth disadvantage vs cells lacking *gsf2* (Fig 2A). They then show that, in a mixed culture of flocculating and non-flocculating cells, flocculating cells preferentially stick to one another with a clever flow cytometry assay. Next, they show that the ratio of *gsf2*IE cells is lower than expected when grown in a co-culture with non-flocculating *gsf2*-deletion cells under conditions that would cause flocculation. They argue that this shows that *gsf2*-deletion cells therefore impose a fitness cost of *gsf2*IE cells and could be considered as cheater cells. They identify conditions in which *gsf2* induced flocculation can improve survival including acetic acid, hydrogen peroxide, and ethanol.

The manuscript then shifts gears to talk about the variability of the *gsf2* gene in both lab strains and natural populations. The authors use long read sequencing to highlight the differences in the number of the different repeats in their lab strain relative to the reference genome, and extract the *gsf2* sequence from existing long read sequencing data from a variety of *S. pombe* isolates, illustrating the wide variability of repeat regions across natural isolates.

The authors then attempt to show that variability of this *gsf2* gene influences flocculation phenotypes. They compare two lab strains (WT and WT_A) with similar backgrounds, but with different numbers of copies of repeat region A in the *gsf2* gene. They claim that these two strains have different flocculation intensities with WT, which has only 16 repeats, showing higher flocculation intensity than WT_A, which has 34 repeats. They make this argument based on three lines of evidence. Most directly, they suggest that the flocculation appears stronger in the WT background (Fig 6F), though as I suggest below this is not very convincing. Secondly they show that, despite showing no difference in growth rate under conditions that would not induce flocculation (Fig 6B), when grown in co-culture, the percentage of cells in the floc from background A increases over time (Fig 6C). Finally, they show that, in flocculation inducing conditions (acetic acid which), the strain with the WT background has a better survival rate than the strain with the WT_A background (Fig 6G).

While there were some compelling findings clearly illustrating triggers and advantages of flocculation and highlighting how variable the *gsf2* gene was, there were some holes in the authors arguments. In particular, the fact that flocculation caused a disadvantage under normal growth conditions was not fully addressed, and the link between variation in repeat regions and flocculation strength was not fully convincing.

While the limited conservation of the *gsf2* gene (restricted to just *S. pombe*) inherently limits the broader impact of the work to some extent, the authors use this example to illustrate the evolutionary concept of green beard genes which is of broader impact.

Major Issues

The extent and nature of the advantage that *gsf2* driven flocculation provides is still unclear. First, as the authors point out in Fig 3A, cells that do not flocculate have a growth advantage over cells that flocculate, however the authors show that in 1% acetic acid, cells that flocculate have an increased survival rate (Fig 3J and 3L). It would seem important to measure growth rate between flocculating and non-flocculating cells in the presence of acetic acid so that we can see if the advantage they gain in survival counterbalances the disadvantage in normal conditions.

While figure 4B shows clearly that the non-flocculating *gsf2*-deletion mutants are not generally included in the large floc created by the *gsf2*IE cells, figure 4A is much less clear. This image in particular doesn't seem to demonstrate that the "spatial organization suggests that the formation of flocs is driven by preferential adhesion of *gsf2*-expressing cells to one another." (Lines 137-139). There is a big clump of cells in the center that seems to include as many GFP labeled *gsf2*-delete planktonic cells as it does mCherry cells. Perhaps there are more/bigger small bright *gsf2*-delete GFP clumps of cells in the middle z-level (54um) than there are outside of the clump for WT flocculating mCherry cells, but the majority of the GFP labeled cells seem to be part of the large floc. Furthermore, these are just single fields of view that are not quantified in any way. For such a subtle effect, the authors should quantify this observation and show it is true for more than one image, and preferably with biological replicates.

In figure 3H the authors argue that flocculating cells suffer when in co-culture with non-flocculating cells. This figure lacks a control that shows that the initial mix was 1:1 (i.e. from time 0). In line 67 it says there is a significant reduction, but I assume this means a reduction from the ideal 50% one would expect if it were a 1:1 mix of strains. It is also strange how the percentage dips even more at 60 hours and then increases again at 72 hours. If this is just variability of the assay it argues that more replicates

should be obtained.

The evidence that variability in the repeat regions influences the ability to flocculate is not convincing. In line 213, the authors say that "the flocculation intensity of gsf2IE appears to be stronger than that of gsf2AIE (Fig 6F)" but this is a very subjective assessment. To make this claim the authors should attempt to quantify flocculation using those images, or else use a less subjective assay, such as filtering or centrifugation to assess the level of flocculation.

The percentage of gsf2IE in the floc is increasing relative to gsf2AIE over time (Fig 6C), however could this be because the overall amount of gsf2AIE decreases over that time frame in EMM media which induces flocculation? It is clear that this is not the case in YE media (Fig 6B), however the EMM is a very different media. They would need to show that the percentage of gsf2IE also goes down in the supernatant.

Finally, while the evidence that gsf2IE is more resistant to acetic acid than gsf2AIE is convincing (Fig 6G), and that this is gsf2 specific and not due to some other effect of the background (Fig 6E), it is possible that this may be due to an effect of gsf2 that is not related to flocculation.

It is likely to be of significant interest to the rest of the *S. pombe* community that the authors had two strains with similar backgrounds in their lab in which there was such a large difference the gsf2 gene, especially since it contributes to variable phenotypes for resistance to stresses like acids and oxidative stress and flocculation. The authors discuss more detail about these strains in the methods, but this information should be emphasized in the main text.

Furthermore, with repetitive and highly variable regions in the genome there is a worry that any transformation could trigger expansions or contractions in the number of repeat regions. Can the authors do something (colony PCR, targeted sanger sequencing, etc) to reassure the reader that the gsf2IE and gsf2AIE strains retained similar repeat patterns to their parental strains (WT and WT_A)?

The authors describe four repeat regions in Figure 5 and line 206, however they do not make clear how they define these four regions. Regions A and B are defined in Matsuzawa et al 2011. The authors should note that they have been previously defined in that work. That work does not define regions C and D. If the authors did not define those regions in this work, they should cite the work where those regions are defined.

The authors show that gsf2 is important for flocculation in response to acetic acid stress, however, expression of gsf2 drops after four hours of exposure to acetic acid, even though cells still flocculate. This seems counterintuitive and the authors do not attempt to explain this. Can the authors speculate on whether the gsf2 protein is still likely to be present after that timeframe, or whether gsf2 is important for initiating flocculation, but is less important for flocculation as time goes on.

Overall I found the individual sections generally clear, however the overall flow of the manuscript was disjointed. For example,

The discussion of the variation in the gsf2 gene began in figure 1A and then was not mentioned again until figure 5.

The microscopy images illustrating flocculation came as a separate figure 4 which was presented before most of the results from figure 3 were presented.

There was often little explanation linking sections and motivating specific experiments. For instance, for figure 3F, it is not clear why the authors chose to mix the cells in that experiment at a ratio of 100:1 rather than 1:1.

The explanation for why gsf2 delete cells might be thought of as cheaters (line 61) comes after they are first described as cheaters (line 43)

Also there were a number of grammar issues and some problems with references (see minor comments below) that indicate that the authors will need to do some significant proofreading.

Minor comments:

M1) Reference issues (there may be more that I didn't notice):

Darwin 2016 in opening paragraph

Two identical references for Matsuzawa et al 2013

Kumar et al should be Kwon et al <https://pubmed.ncbi.nlm.nih.gov/23236291/>

M2) Figure 1 shows a lot of raw data that is repetitive. To make it more concise, the authors could use representative images and show all the replicates in the supplement. It would also be helpful to combine some of the relative expression barplots to one figure and use the same y-axis. This would also illustrate the stark decrease in gsf2 expression over time.

M3) The legend for Figure 1 should mention the assay used to measure expression (QPCR). Also for the expression bar plots it is not clear what the error bars represent.

M4) I assume all repeats are biological repeats (i.e. from separate starting cultures), rather than technical replicates which is

good, however if these are technical replicates (repeated samples from the same starting culture), that should be mentioned in the methods.

M5) Figure 2 could also be made more concise by using representative images and plotting all barplots on the same axis.

M6) The reference for Figure 3J should be near line 178.

M7) Line 183 refers to "all four stresses", but the authors only describe three stresses.

M8) For figure 3F, at 48 hours there is no significant difference between normal conditions and in acetic acid. Can the authors explain why this might occur? If this is due to normal variability of the assay it indicates that there may need to be more replicates to get a representative sample.

M9) In line 45, it seems like *gsf2+* refers to *gsf2* IE cells based on the figure - The authors should use consistent terminology.

M10) In Fig 3C: The numbers appear to be p-values from a t-test, but it would be good to specify this in the caption.

M11) For figures 3C, D and E, it would be easier to interpret if all of them used the same axes.

M12) Fig 3G: Indicate in the figure where the samples are collected

M13) For Fig 3H, nothing is listed in the figure caption.

M14) The key for figure 5A is in an awkward location. It should be made more clear that it applies to that figure and not figure 5C which it is directly below. Also that figure would be easier to follow if the columns with the number of repeat regions used the same color scheme as the regions themselves.

M15) Shouldn't line 195 read A 5 to 72? Also shouldn't Fig 5C have a point for JB1206 which has 72 A regions, or are those strains with premature stop codons filtered out of further analysis? If this is the case, that should be mentioned explicitly.

M16) The evidence for no homologous gene being identified in other closely related species (Fig S4) is that the gene doesn't appear in the same location as would be expected by synteny. While this is suggestive, I feel like a few more things ought to be checked to make this claim. For instance, are the individual repetitive regions conserved in those other species or else more widely? I attempted a quick tblastN search of repetitive region A in those other *Schizosaccharomyces* species and across the whole NCBI genome database general database and didn't find any hits, but it might be informative to add a similar but more thorough search to back this claim up.

M17) Figure 6B shows that there is no difference in growth between *gsf2*IE and *gsf2*AIE. Why not show that there is a difference in growth when flocculation is induced? The competition assay is somewhat convincing, but it would be useful to know if this is an effect that is based on the behavior in a monoculture or only manifests in a competitive assay.

M18) Line 262 error "*gfs2*"

M19) It would be easier to follow if the reader did not need to rely so heavily on the figure captions to interpret the figures. For instance, for figure 6C, the y-axis is hard to interpret from the figure alone; it is not clear from the figure itself that the assay was a competition between *gsf2*IE and *gsf2*AIE.

Referee #2:

Wu & Han EMBO

This is an interesting paper showing results analogous to previous work on FLO1 in another yeast species; but with an independent evolution - this time *gsf2* in *S. pombe*. While the results are not new in some sense, the discovery of an independent evolution is very interesting to see. This paper examines: (i + ii) the role of *gsf2* and acid stress in flocculation; (iii) cost of suppressing *gsf2*; (iv) preferential adherence; (v) relative fitness under stress; (vi) how flocculation defends against stress; (vii) variability of *gsf2*.

Specific comments / questions:

1. Page 8-9. Isn't it key to show that different variants of *gsf2* preferentially flocculate with themselves and not the other variants? i.e. that there are multiple colours to the greenbeard?
2. Re "*gsf2*-expressing cells preferentially adhere to each other". What is the null model? What proportion of incorporated mutants would imply preferential or not?

3. Page 7-8. A fitness cost of cheats is expected even without preferential adherence etc. <https://academic.oup.com/jeb/article-abstract/23/4/738/7324583?login=false>

4. Can you show that it is functioning as a green beard and not just a genetic cue for kin recognition (e.g. <https://www.nature.com/articles/s41467-022-31545-4> or <https://www.pnas.org/doi/abs/10.1073/pnas.2220761120>)?

As is my policy, I waive anonymity
Stu West 10 December 2025

Response to Reviewers' Comments

Referee #1:

The authors show that cells treated with acid flocculate, that this flocculation is correlated with *gsf2* expression and that this flocculation is dependent on the presence of *gsf2* by showing that it is absent in a *gsf2* deletion and present when *gsf2* is overexpressed, even in the absence of acetic acid. The authors then show that cells flocculating due to *gsf2* overexpression show a growth disadvantage vs cells lacking *gsf2* (Fig 2A). They then show that, in a mixed culture of flocculating and non-flocculating cells, flocculating cells preferentially stick to one another with a clever flow cytometry assay. Next, they show that the ratio of *gsf2*IE cells is lower than expected when grown in a co-culture with non-flocculating *gsf2*-deletion cells under conditions that would cause flocculation. They argue that this shows that *gsf2*-deletion cells therefore impose a fitness cost of *gsf2*IE cells and could be considered as cheater cells. They identify conditions in which *gsf2* induced flocculation can improve survival including acetic acid, hydrogen peroxide, and ethanol.

The manuscript then shifts gears to talk about the variability of the *gsf2* gene in both lab strains and natural populations. The authors use long read sequencing to highlight the differences in the number of the different repeats in their lab strain relative to the reference genome, and extract the *gsf2* sequence from existing long read sequencing data from a variety of *S. pombe* isolates, illustrating the wide variability of repeat regions across natural isolates.

The authors then attempt to show that variability of this *gsf2* gene influences flocculation phenotypes. They compare two lab strains (WT and WT_A) with similar backgrounds, but with different numbers of copies of repeat region A in the *gsf2* gene. They claim that these two strains have different flocculation intensities with WT, which has only 16 repeats, showing higher flocculation intensity than WT_A, which has 34 repeats. They make this argument based on three lines of evidence. Most directly, they suggest that the flocculation appears stronger in the WT background (Fig 6F), though as I suggest below this is not very convincing. Secondly they show that, despite showing no difference in growth rate under conditions that would not induce flocculation (Fig 6B), when grown in co-culture, the percentage of cells in the floc from background A increases over time (Fig 6C). Finally, they show that, in flocculation inducing conditions (acetic acid which), the strain with the WT background has a better survival rate than the strain with the WT_A background (Fig 6G).

While there were some compelling findings clearly illustrating triggers and advantages of flocculation and highlighting how variable the *gsf2* gene was, there were some holes in the authors arguments. In particular, the fact that flocculation caused a disadvantage under normal growth conditions was not fully addressed, and the link between variation in repeat regions and flocculation strength was not fully convincing.

While the limited conservation of the *gsf2* gene (restricted to just *S. pombe*) inherently limits the broader impact of the work to some extent, the authors use this example to illustrate the evolutionary concept of green beard genes which is of broader impact.

Response: Thanks so much for the nice summary! Please see the following for our detailed point-to-point response.

Major Issues

(1) The extent and nature of the advantage that *gsf2* driven flocculation provides is still unclear. First, as the authors point out in Fig 3A, cells that do not flocculate have a growth advantage over cells that flocculate, however the authors show that in 1% acetic acid, cells that flocculate have an increased survival rate (Fig 3J and 3L). It would seem important to measure growth rate between flocculating and non-flocculating cells in the presence of acetic acid so that we can see if the advantage they gain in survival counterbalances the disadvantage in normal conditions.

Response: Thanks for this excellent comment! As suggested, we directly measured the growth of flocculating (*gsf2IE*) cells and non-flocculating (*gsf2Δ*) cells under acetic acid stress. Indeed, we found significantly higher growth rate of *gsf2IE* cells compared to *gsf2Δ* cells under acetic acid stress. The results have been provided as Fig EV1.

(2) While figure 4B shows clearly that the non-flocculating *gsf2*-deletion mutants are not generally included in the large floc created by the *gsf2IE* cells, figure 4A is much less clear. This image in particular doesn't seem to demonstrate that the "spatial organization suggests that the formation of flocs is driven by preferential adhesion of *gsf2*-expressing cells to one another." (Lines 137-139). There is a big clump of cells in the center that seems to include as many GFP labeled *gsf2*-delete planktonic cells as it does mCherry cells. Perhaps there are more/bigger small bright *gsf2*-delete GFP clumps of cells in the middle z-level (54um) than there are outside of the clump for WT flocculating mCherry cells, but the majority of the GFP labeled cells seem to be part of the large floc. Furthermore, these are just single fields of view that are not quantified in any way. For such a subtle effect, the authors should quantify this observation and show it is true for more than one image, and preferably with biological replicates.

Response: Thanks for this insightful critique. The flocs in the original Fig. 4A were formed by WT cells under acetic acid stress, which induced the expression of native *gsf2* in WT cells. The flocs in Fig. 4B were formed by *gsf2IE* cells, in which the expression of *gsf2* was driven by the *nmt41* promoter in EMM medium. This fundamental difference in expression pattern might explain the difference in resulting floc architecture, and the floc architecture appears to be more pronounced in Fig. 4B. To address the reviewer's concerns, we performed additional confocal imaging across multiple independent fields of view. These images, now compiled in Supplementary Figure S5, consistently demonstrate the preferential adhesion of *gsf2*-expressing cells and reinforce the reproducibility of the floc architectures under both expression conditions.

(3) In figure 3H the authors argue that flocculating cells suffer when in co-culture with non-flocculating cells. This figure lacks a control that shows that the initial mix was 1:1 (i.e. from time 0). In line 67 it says there is a significant reduction, but I assume this means a reduction from the ideal 50% one would expect if it were a 1:1 mix of strains. It is also strange how the percentage dips even more at 60 hours and then increases again at 72 hours. If this is just variability of the assay it argues that more replicates should be obtained.

Response: As suggested, we added the time-zero (T0) data point and increased the number of biological replicates to five. These revisions have resolved the concerns raised.

(4) The evidence that variability in the repeat regions influences the ability to flocculate is not convincing. In line 213, the authors say that "the flocculation intensity of *gsf2IE* appears to be stronger than that of *gsf2AIE* (Fig 6F)" but this is a very subjective assessment. To make this claim the authors should attempt to quantify flocculation using those images, or else use a less subjective assay, such as filtering or centrifugation to assess the level of flocculation.

Response: Thanks for this suggestion. However, to quantify the flocculation intensity, we encountered a fundamental technical challenge due to the biological material itself: the flocs formed in our system are inherently loose, fragile, and prone to disaggregation during physical

manipulation (e.g., filtration, centrifugation). This instability made consistent and reliable quantification highly challenging. We added more replicates and observed a consistent qualitative difference (Fig. 6E; Appendix Figure S7A).

(5) The percentage of *gsf2IE* in the floc is increasing relative to *gsf2AIE* over time (Fig 6C), however could this be because the overall amount of *gsf2AIE* decreases over that time frame in EMM media which induces flocculation? It is clear that this is not the case in YE media (Fig 6B), however the EMM is a very different media. They would need to show that the percentage of *gsf2IE* also goes down in the supernatant.

Response: As suggested, we conducted the experiment measuring the percentage of *gsf2^AIE* in the supernatant. The results are now included in Fig. 6D. As expected, the percentage of *gsf2IE* went down in the supernatant.

Finally, while the evidence that *gsf2IE* is more resistant to acetic acid than *gsf2AIE* is convincing (Fig 6G), and that this is *gsf2* specific and not due to some other effect of the background (Fig 6E), it is possible that this may be due to an effect of *gsf2* that is not related to flocculation. It is likely to be of significant interest to the rest of the *S. pombe* community that the authors had two strains with similar backgrounds in their lab in which there was such a large difference the *gsf2* gene, especially since it contributes to variable phenotypes for resistance to stresses like acids and oxidative stress and flocculation. The authors discuss more detail about these strains in the methods, but this information should be emphasized in the main text.

Response: We agree! We added the key description of the *gsf2IE* and *gsf2AIE* strains in the Results section.

Furthermore, with repetitive and highly variable regions in the genome there is a worry that any transformation could trigger expansions or contractions in the number of repeat regions. Can the authors do something (colony PCR, targeted sanger sequencing, etc) to

reassure the reader that the *gsf2IE* and *gsf2AIE* strains retained similar repeat patterns to their parental strains (WT and WT_A)?

Response: Thanks for this excellent point. We acknowledge that complex, variable repeat regions could potentially undergo unintended expansion or contraction during transformation. To address this point directly, we performed experiments to verify the *gsf2* gene in our constructed strains (*gsf2IE* and *gsf2^AIE*, each with three biological replicates, R1–R3). Given the considerable length and repetitive nature of the *gsf2* sequence, we performed verification using a two-stage, segmental PCR amplification strategy (as indicated in the following figure). Two pairs of primers were designed based on non-repetitive regions: P1/P2 amplify Part1, and P3/P4 amplify Part2. Based on sequence information, the expected sizes are as follows: For the WT^A background: Part1 = 8604 bp, Part2 = 2369 bp. For the WT background: Part1 = 4390 bp, Part2 = 2369 bp. PCR results (Figure B) show that the amplified fragment sizes for all constructed strains match the predicted sizes of their corresponding parental strains, and the three biological replicates yield consistent banding patterns. These results demonstrate that *gsf2IE* and *gsf2^AIE* retained the same *gsf2* length as their parental strains after transformation, with no detectable expansion or contraction of repetitive units.

The authors describe four repeat regions in Figure 5 and line 206, however they do not make clear how they define these four regions. Regions A and B are defined in Matsuzawa et al 2011. The authors should note that they have been previously defined in that work. That work does not define regions C and D. If the authors did not define those regions in this work, they should cite the work where those regions are defined.

Response: As suggested, we revised the manuscript to explicitly clarify the origin of each repeat region definition. In the revised text, we now stated: “Gsf2 proteins possess four repeat motifs, designated A to D. Motifs A and B have been defined previously (Matsuzawa et al., 2011), and motifs C and D are newly defined in this study based on our sequence analysis.”

The authors show that *gsf2* is important for flocculation in response to acetic acid stress, however, expression of *gsf2* drops after four hours of exposure to acetic acid, even though cells still flocculate. This seems counterintuitive and the authors do not attempt to explain

this. Can the authors speculate on whether the *gsf2* protein is still likely to be present after that timeframe, or whether *gsf2* is important for initiating flocculation, but is less important for flocculation as time goes on.

Response: Thanks for this excellent point. We suspect that Gsf2 protein is still like present after that timeframe, given *gsf2* is still expressed. We added a note on this in the revised manuscript.

Overall I found the individual sections generally clear, however the overall flow of the manuscript was disjointed. For example, The discussion of the variation in the *gsf2* gene began in figure 1A and then was not mentioned again until figure 5. The microscopy images illustrating flocculation came as a separate figure 4 which was presented before most of the results from figure 3 were presented.

Response: Overall, thanks for all these comments on writing. We have revised the manuscript based on this reviewer's suggestion. Regarding the order of figures, we moved the microscopy images (previously Figure 4) to follow the results of Figure 3. Concerning the timing of the discussion on the *gsf2* gene variation, we would like to clarify our aims: the purpose of presenting the variant in Figure 1A was to clarify that the wild-type strain used in this study carries an inherent sequence difference in the *gsf2* gene compared to the standard reference strain (972h-). This serves as a clarification of background strain, rather than an intention to discuss the extent of variation in the gene at that point. It is in Figure 5 that we focus on exploring the high natural polymorphism of this gene within population.

There was often little explanation linking sections and motivating specific experiments. For instance, for figure 3F, it is not clear why the authors chose to mix the cells in that experiment at a ratio of 100:1 rather than 1:1.

Response: We used a ~100:1 starting ratio to model the invasion of rare non-cooperative cells (*gsf2*Δ) into a cooperative population (*gsf2IE*). This tests whether cooperation (flocculation) provides a defense against cheaters. We added this explanation in the revised text.

The explanation for why *gsf2* delete cells might be thought of as cheaters (line 61) comes after they are first described as cheaters (line 43)

Response: There might be line number problems. Nevertheless, we understood the problem and reversed the order of the sections “flocculation escapes cheater exploitation” and “*gsf2*Δ imposes a fitness cost on *gsf2IE*”. Now we defined and tested cheaters first.

Also there were a number of grammar issues and some problems with references (see minor comments below) that indicate that the authors will need to do some significant proofreading.

Response: Thanks! We conducted a thorough proofreading and language polishing of the entire manuscript to ensure clarity, accuracy, and adherence to academic standards.

Minor comments:

M1) Reference issues (there may be more that I didn't notice):

Darwin 2016 in opening paragraph

Two identical references for Matsuzawa et al 2013

Kumar et al should be Kwon et al <https://pubmed.ncbi.nlm.nih.gov/23236291/>

Response: Corrected! We also checked and corrected other reference issues in the revised manuscript.

M2) Figure 1 shows a lot of raw data that is repetitive. To make it more concise, the authors could use representative images and show all the replicates in the supplement. It would also be helpful to combine some of the relative expression barplots to one figure and use the same y-axis. This would also illustrate the stark decrease in *gsf2* expression over time.

Response: We revised Figure 1 by selecting representative images for the main figure and moving the complete replicate data to the supplementary materials. This has made Figure 1 more concise and visually focused.

M3) The legend for Figure 1 should mention the assay used to measure expression (QPCR). Also for the expression bar plots it is not clear what the error bars represent.

Response: The legend for Figure 1 has been revised to specify that expression was measured by qPCR and to clarify that error bars represent the mean \pm SD of three biological replicates.

M4) I assume all repeats are biological repeats (i.e. from separate starting cultures), rather than technical replicates which is good, however if these are technical replicates (repeated samples from the same starting culture), that should be mentioned in the methods.

Response: As suggested, we have now explicitly stated in the Methods section (under “Statistical Analysis”) that all replicates reported in this study are independent biological replicates, derived from separate starting cultures.

M5) Figure 2 could also be made more concise by using representative images and plotting all barplots on the same axis.

Response: Done as suggested!

M6) The reference for Figure 3J should be near line 178.

Response: Done as suggested!

M7) Line 183 refers to "all four stresses", but the authors only describe three stresses.

Response: Corrected!

M8) For figure 3F, at 48 hours there is no significant difference between normal conditions and in acetic acid. Can the authors explain why this might occur? If this is due to normal variability of the assay it indicates that there may need to be more replicates to get a representative sample.

Response: Thanks! To address this concern and ensure the robustness of the conclusions, we increased the number of independent biological replicates from three to five for this competitive assay.

M9) In line 145, it seems like gsf2+ refers to gsf2IE cells based on the figure-The authors should use consistent terminology.

Response: Corrected!

M10) In Fig 3C: The numbers appear to be p-values from a t-test, but it would be good to specify this in the caption.

Response: The figure caption has been updated to specify that the indicated values are p-values from Student's t-tests.

M11) For figures 3C, D and E, it would be easier to interpret if all of them used the same axes.

Response: The axes in Figures 3C, D, and E have been standardized as suggested.

M12) Fig 3G: Indicate in the figure where the samples are collected

Response: Done as suggested!

M13) For Fig 3H, nothing is listed in the figure caption.

Response: The figure caption for Fig. 3H has now been updated in the revised manuscript.

M14) The key for figure 5A is in an awkward location. It should be made more clear that it applies to that figure and not figure 5C which it is directly below. Also that figure would be easier to follow if the columns with the number of repeat regions used the same color scheme as the regions themselves.

Response: Done as suggested!

M15) Shouldn't line 195 read A 5 to 72? Also shouldn't Fig 5C have a point for JB1206 which has 72 A regions, or are those strains with premature stop codons filtered out of further analysis? If this is the case, that should be mentioned explicitly.

Response: The statement on line 195 regarding A repeats (5-35) is accurate, as it specifically describes the range within the filtered set of strains carrying intact *gsf2* ORFs used for subsequent analysis. We clarified this in the revised manuscript. The absence of strain JB1206 (with 72 A repeats) from Fig. 5C is consistent with our analytical framework, as strains harboring premature stop codons (likely pseudogenes) in *gsf2* were excluded from this analysis.

M16) The evidence for no homologous gene being identified in other closely related species (Fig S4) is that the gene doesn't appear in the same location as would be expected by synteny. While this is suggestive, I feel like a few more things ought to be checked to make this claim. For instance, are the individual repetitive regions conserved in those other species or else more widely? I attempted a quick tblastN search of repetitive region A in those other Schizosaccharomyces species and across the whole NCBI genome database general database and didn't find any hits, but it might be informative to add a similar but more thorough search to back this claim up.

Response: We conducted the suggested tBLASTn search using the individual repetitive motifs (A–C). The results confirm the absence of significant homologous sequences outside of *S. pombe*, and this result has been added to the revised manuscript.

M17) Figure 6B shows that there is no difference in growth between *gsf2IE* and *gsf2AIE*. Why not show that there is a difference in growth when flocculation is induced? The competition assay is somewhat convincing, but it would be useful to know if this is an effect that is based on the behavior in a monoculture or only manifests in a competitive assay.

Response: We performed the experiment suggested by the reviewer: we compared the growth of *gsf2IE* and *gsf2^AIE* strains under flocculation-inducing conditions and found slower growth for *gsf^A2IE*. The results were presented in Fig. EV2.

M18) Line 262 error "gfs2"

Response: Corrected!

M19) It would be easier to follow if the reader did not need to rely so heavily on the figure captions to interpret the figures. For instance, for figure 6C, the y-axis is hard to interpret from the figure alone; it is not clear from the figure itself that the assay was a competition between *gsf2IE* and *gsf2AIE*.

Response: We revised Figure 6C by updating the Y-axis title to "Percentage of *gsf2IE* (in competition with *gsf2^AIE*)" to explicitly indicate the competitive assay.

Referee #2:

Wu & Han EMBO

This is an interesting paper showing results analogous to previous work on FLO1 in another yeast species; but with an independent evolution - this time *gsf2* in *S. pombe*. While the results are not new in some sense, the discovery of an independent evolution is very interesting to see. This paper examines: (i + ii) the role of *gsf2* and acid stress in flocculation; (iii) cost of suppressing *gsf2*; (iv) preferential adherence; (v) relative fitness under stress; (vi) how flocculation defends against stress; (vii) variability of *gsf2*.

Response: Thanks so much for the nice summary!

Specific comments / questions:

1. Page 8-9. Isn't it key to show that different variants of *gsf2* preferentially flocculate with themselves and not the other variants? i.e. that there are multiple colours to the greenbeard?

Response: Thanks for the comment on "multicolored greenbeard" effect. *Gsf2* recognizes and binds to galactose residues on extracellular sugar chains. We found that different natural *gsf2* variants (e.g., *gsf2^{AIE}* versus *gsf2IE*) confer different flocculation intensities, which ultimately lead to divergent fitness outcomes under stress (Fig. 6F). More importantly, in competition experiments using mixed cultures, the *gsf2IE* variant, which possesses stronger flocculation ability, is preferentially recruited and enriched within the flocculent fraction (Fig. 6C and D; Appendix Figure S7). These results suggest that between different variants, there may be a non-random, performance-based social assortment dependent on their functional strength (i.e., flocculation "efficacy"), rather than an absolute, specific "color" matching.

2. Re "*gsf2*-expressing cells preferentially adhere to each other". What is the null model? What proportion of incorporated mutants would imply preferential or not?

Response: The null hypothesis we employed is that incorporation into flocs is a random process independent of a cell's *gsf2* expression status. Specifically, in a 1:1 initial mixture of *gsf2IE* and *gsf2Δ*, the expected proportion of *gsf2IE* cells within flocs is 50%. To test this, we directly measured the proportions of *gsf2IE* cells over time in both the supernatant and the flocculent fraction. Our data show a statistically significant and consistent decrease in the supernatant coupled with a corresponding increase within the flocs. These results reject the null hypothesis and provides strong evidence for the preferential, non-random recruitment of *gsf2IE* cells into flocs. We clarified this in the revised manuscript.

3. Page 7-8. A fitness cost of cheats is expected even without preferential adherence etc. <https://academic.oup.com/jeb/article-abstract/23/4/738/7324583?login=false>

Response: Thanks for highlighting the relevant reference (Jiricny et al., 2010). We agree that a fitness cost imposed by cheaters on cooperators is a fundamental expectation in social evolution. In the revised manuscript, we integrated this into our discussion. We cited Jiricny et al. (2010) and went further. We demonstrated that despite imposing a fitness cost, the *gsf2Δ* cheater cannot

invade the population, as its frequency within flocs decreases over time. This suggests that the *gsf2* system can escape cheater exploration.

4. Can you show that it is functioning as a green beard and not just a genetic cue for kin recognition

(e.g. <https://www.nature.com/articles/s41467-022-31545-4> or <https://www.pnas.org/doi/abs/10.1073/pnas.2220761120>)?

Response: Thanks for the comments. Because the extremely long and repetitive nature of *gsf2* gene, it is technically unfeasible to clone the gene and transfer it into other genetic backgrounds. However, except the *gsf2* gene, the *gsf2IE* and the *gsf2Δ* strains share genetic relatedness of 1, because they are essentially derived from the same WT strain. *gsf2*-expressing cells “recognize” and preferentially adhere to other *gsf2*-expressing cells. Therefore, we think it is a green beard rather than a genetic cue for kin recognition.

Dear Dr. Han

Thank you for the submission of your revised manuscript to our offices. We have now received the enclosed reports from the referees that were asked to assess it. EMBOR-2025-62960V2 still has minor comments that I would like you to address before we can make a final decision on your manuscript. Please provide a point-by-point response letter for addressing these comments.

Furthermore, our editorial assistants have made several important comments regarding the more technical aspects of your paper. These must be corrected in the next revision. Please note you do not need to address those in your point-by-point response letter, but just made the edits according to the assistants' comments.

Yehu Moran
Academic Editor
EMBO Reports

****Editorial assistants' comments****

AC/CRedit: need to be removed from the manuscript text and appear only in the submission system.

FUNDING INFO: OK, but the separate 'Funding' title is not needed since this info goes under Acknowledgments; no Acknowledgments text provided which is unusual, please check and make sure.

FIGURE CALLOUTS: Table S2, need to be corrected to Appendix Table S2

SYNOPSIS IMAGE: missing, please provide.

SYNOPSIS TEXT: missing, please provide.

R&T TABLE: in the manuscript text, but also provide separately; the one in the manuscript needs to be removed.

SOURCE DATA (SD): SD provided with completed checklist; discrepancies: 1E provided, but 1G missing even though it has been checked; pls check whether the micr. images have been deposited in the BioImage Archive; each SD folder needs to be upldd separately

==>Discrepancy resolved; SD folders for main and EV figure need to be upldd separately while the ones for the Appendix Figures can be combined into one folder.

EXTRA NOTES:

- 'Expanded View for this article is available online.' should be removed from the manuscript text.
- Materials and Methods should be renamed Methods

- Appendix Figure S5. Please supply the source data for this figure, as the microscopy images are pixelated under image analysis.

Figure Legends - Comments

- Please indicate the statistical test used for data analysis in the legend of figure 6F

- Please note that the box plots need to be defined in terms of minima, maxima, centre, bounds of box and whiskers, and percentile in the legend of figure 5C

- Please note that information related to n is missing in the legends of figures 3B, 5C, 6F

- Please note that the error bars are not defined in the legends of figures 3B, 6F

****Referee comments****

Referee #1:

The authors have strengthened the manuscript with additional experiments and clarifications and almost all my concerns and the concerns of the other reviewer have been addressed. I have three remaining comments:

1) I still don't find the images in figure 4A and Figure S5 for the coculture of WT-mCherry and gsf2del-GFP in 0.1% Acetic Acide to support the interpretation in lines 190-200:

"We investigated the spatial arrangement of mixed cell populations using confocal microscopy. WT cells (labeled with mCherry) and gsf2Δ cells (labeled with GFP) were co-cultured at an approximate 1:1 ratio under 0.1% acetic acid induced gsf2 expression (Fig 4A; Appendix Fig S5A for more views). The results demonstrated that most WT cells aggregated, forming flocs. Although a large fraction of gsf2Δ cells were planktonic and external to the flocs, some were incorporated into the aggregates and intermixed with the WT cells. This spatial organization suggests that the formation of flocs is driven by the preferential adhesion of gsf2-expressing cells to one another. Moreover, a similar phenomenon was observed in the co-culture of gsf2IE (mCherry-labeled) and gsf2Δ (GFP-labeled) cells at an equal ratio (Fig 4B; Appendix Fig S5B for more views). These results suggest gsf2-expressing cells preferentially adhere to each other."

It is hard to say whether there are more planktonic *gsf2*-del-GFP cells than there are planktonic WT-mCherry cells. I would suggest the authors adjust their wording to the following:

"We investigated the spatial arrangement of mixed cell populations using confocal microscopy. WT cells (labeled with mCherry) and *gsf2* Δ cells (labeled with GFP) were co-cultured at an approximate 1:1 ratio under 0.1% acetic acid induced *gsf2* expression (Fig 4A; Appendix Fig S5A for more views). The results demonstrated that most WT cells aggregated, forming flocs, and that many *gsf2* Δ cells were incorporated into the aggregates and intermixed with the WT cells. For the co-culture of *gsf2*IE (mCherry-labeled) and *gsf2* Δ (GFP-labeled) cells inoculated at an equal ratio (Fig 4B; Appendix Fig S5B for more views), a large fraction of *gsf2* Δ cells were planktonic and external to the flocs suggesting that the formation of flocs is driven by the preferential adhesion of *gsf2*-expressing cells to one another."

Whether or not the authors agree to this suggestion, the fact that they have multiple replicate images will allow the reader to make up their own mind.

2) Similarly in lines 233-234 the authors say

"Consistently, the flocculation intensity of *gsf2*IE appears to be stronger than that of *gsf2*A234 IE (Fig 6E; Appendix Fig S7A for replicates)."

I do not see a strong qualitative difference in figure 6E between the flocculation intensity for *gsf2*AIE and *gsf2*IE. I would argue that the authors should remove that point. However if they would like to keep that point, they have several replicates and readers can make up their own minds.

3) Finally, I don't find the author's response to my concern about possible repeat expansion in the *gsf2*AIE transformed strains relative to the parental strains convincing.

For *gsf2*AIE Part 1 there is hardly any band. Perhaps there is a very faint band somewhere around 4KB (the incorrect size) and another very faint band above 8KB. For *gsf2*IE Part 1, all three replicates show a blur, possibly with a slightly brighter band around the correct size. I would say this is hardly conclusive. For part 2 it is all consistent, but the band appears to be in between 3-4kb rather than 2.4kb. Also the authors say the bands "match the predicted sizes of their corresponding parental strains" but the authors do not show the colony PCR of the parental WT or WT_A backgrounds which would seem to be the obvious control. I would argue that the authors should state in the methods that they checked the *gsf2*AIE and *gsf2*IE strains for evidence of repeat expansion following transformation, and three replicates had repeatable band structures and show this experiment along in the appendix so readers can decide for themselves whether it is convincing that these transformants are representative of their parental lineages.

Referee #2:

The authors have done a good job of addressing my points and I have no major comments. My only comment is re their point 4 - yes that is how the strains were constructed for laboratory experiments, but the key question is re what happens in natural populations.

Response to Reviewers' Comments

Referee #1:

The authors have strengthened the manuscript with additional experiments and clarifications and almost all my concerns and the concerns of the other reviewer have been addressed. I have three remaining comments:

1) I still don't find the images in figure 4A and Figure S5 for the coculture of WT-mCherry and *gsf2*del-GFP in 0.1% Acetic Acid to support the interpretation in lines 190-200: "We investigated the spatial arrangement of mixed cell populations using confocal microscopy. WT cells (labeled with mCherry) and *gsf2*Δ cells (labeled with GFP) were co-cultured at an approximate 1:1 ratio under 0.1% acetic acid induced *gsf2* expression (Fig 4A; Appendix Fig S5A for more views). The results demonstrated that most WT cells aggregated, forming flocs. Although a large fraction of *gsf2*Δ cells were planktonic and external to the flocs, some were incorporated into the aggregates and intermixed with the WT cells. This spatial organization suggests that the formation of flocs is driven by the preferential adhesion of *gsf2*-expressing cells to one another. Moreover, a similar phenomenon was observed in the co-culture of *gsf2*IE (mCherry-labeled) and *gsf2*Δ (GFP-labeled) cells at an equal ratio (Fig 4B; Appendix Fig S5B for more views). These results suggest *gsf2*-expressing cells preferentially adhere to each other." It is hard to say whether there are more planktonic *gsf2*-del-GFP cells than there are planktonic WT-mCherry cells. I would suggest the authors adjust their wording to the following: "We investigated the spatial arrangement of mixed cell populations using confocal microscopy. WT cells (labeled with mCherry) and *gsf2*Δ cells (labeled with GFP) were co-cultured at an approximate 1:1 ratio under 0.1% acetic acid induced *gsf2* expression (Fig 4A; Appendix Fig S5A for more views). The results demonstrated that most WT cells aggregated, forming flocs, and that many *gsf2*Δ cells were incorporated into the aggregates and intermixed with the WT cells. For the co-culture of *gsf2*IE (mCherry-labeled) and *gsf2*Δ (GFP-labeled) cells inoculated at an equal ratio (Fig 4B; Appendix Fig S5B for more views), a large fraction of *gsf2*Δ cells were planktonic and external to the flocs suggesting that the formation of flocs is driven by the preferential adhesion of *gsf2*-expressing cells to one another." Whether or not the authors agree to this suggestion, the fact that they have multiple replicate images will allow the reader to make up their own mind.

Response: Thank you for the helpful suggestion! We adjusted the interpretation based on this reviewer's comments!

2) Similarly in lines 233-234 the authors say

"Consistently, the flocculation intensity of *gsf2*IE appears to be stronger than that of *gsf2*AIE (Fig 6E; Appendix Fig S7A for replicates)."

I do not see a strong qualitative difference in figure 6E between the flocculation intensity for *gsf2*AIE and *gsf2*IE. I would argue that the authors should remove that point. However if they would like to keep that point, they have several replicates and readers can make up their own minds.

Response: Thank you for the suggestion. We removed the mentioned sentence as suggested.

3) Finally, I don't find the author's response to my concern about possible repeat expansion in the *gsf2AIE* transformed strains relative to the parental strains convincing.

For *gsf2AIE* Part 1 there is hardly any band. Perhaps there is a very faint band somewhere around 4KB (the incorrect size) and another very faint band above 8KB. For *gsf2IE* Part 1, all three replicates show a blur, possibly with a slightly brighter band around the correct size. I would say this is hardly conclusive. For part 2 it is all consistent, but the band appears to be in between 3-4kb rather than 2.4kb. Also the authors say the bands "match the predicted sizes of their corresponding parental strains" but the authors do not show the colony PCR of the parental WT or WT_A backgrounds which would seem to be the obvious control. I would argue that the authors should state in the methods that they checked the *gsf2AIE* and *gsfIE* strains for evidence of repeat expansion following transformation, and three replicates had repeatable band structures and show this experiment along in the appendix so readers can decide for themselves whether it is convincing that these transformants are representative of their parental lineages.

Response: Thanks for this excellent comment. We understand the concern regarding possible repeat expansion in the *gsf2^{AIE}* transformed strains. Due to the high number of repeat sequences, the *gsf2* locus is technically challenging to amplify by PCR. This is particularly true for Part 1, where we frequently observe non-specific bands and a continuous ladder-like pattern, which is a common issue when amplifying repetitive regions. This technical limitation makes it difficult to obtain clean, conclusive bands for comparison. However, we did whole-genome sequencing for both WT and WT^A using long-read sequencing, which can resolve the sequencing of complex regions. The length and repeat structures of *gsf2* were annotated based on long-read sequencing. The review comments will be posted, based on which the readers can decide.

Referee #2:

The authors have done a good job of addressing my points and I have no major comments. My only comment is re their point 4 - yes that is how the strains were constructed for laboratory experiments, but the key question is re what happens in natural populations.

Response: Thanks for your positive feedback! We completely agree that understanding what happens in natural populations is the key question going forward. While our current study focused on laboratory-constructed strains to establish the underlying mechanism under controlled conditions, we fully acknowledge that future work will be needed to explore whether similar phenomena occur in natural isolates and ecological settings.

Guan-Zhu Han
Nanjing Normal University
College of Life Science
1 Wenyuan Rd
Nanjing 210023
China

Dear Dr. Han,

I am very pleased to accept your manuscript for publication in the next available issue of EMBO Reports. Thank you for your contribution to our journal.

You may qualify for financial assistance for your publication charges - either via a Springer Nature fully open access agreement or an EMBO initiative. Check your eligibility: <https://link.springer.com/journal/44319/how-to-publish-with-us>

Yours sincerely,

Yehu Moran
Editor
EMBO Reports

>>> Please note that it is EMBO Reports policy for the transcript of the editorial process (containing referee reports and your response letter) to be published as an online supplement to each paper. If you do NOT want this, you will need to inform the Editorial Office via email immediately. More information is available here: <https://link.springer.com/partners/embo-press/editorial-policies#Peer%20review>